behaviour/ecology/biomechanics

biologging, vertical movement, cost of transport, top predator, overall dynamic body acceleration

**Author for correspondence:**
Samantha Andrzejaczek
e-mail: sandrzejaczek@gmail.com

†Present address: Hopkins Marine Station, Stanford University, Pacific Grove, CA 93950, USA.

# Depth-dependent dive kinematics suggest cost-efficient foraging strategies by tiger sharks

Samantha Andrzejaczek[1,2,†], Adrian C. Gleiss[3], Karissa O. Lear[3], Charitha Pattiaratchi[1], Taylor K. Chapple[4,5] and Mark G. Meekan[2]

[1]Oceans Graduate School and The UWA Oceans Institute, The University of Western Australia, Crawley, Western Australia 6009, Australia
[2]The Australian Institute of Marine Science, Crawley, Western Australia 6009, Australia
[3]Centre for Sustainable Aquatic Ecosystems, Harry Butler Institute, Murdoch University, Murdoch, Western Australia 6150, Australia
[4]Coastal Oregon Marine Experiment Station, Oregon State University, Newport, OR 97365, USA
[5]Hopkins Marine Station, Stanford University, Pacific Grove, CA, 93950, USA

SA, 0000-0002-9929-7312; ACG, 0000-0002-9960-2858; KOL, 0000-0002-2648-8564

Tiger sharks, *Galeocerdo cuvier,* are a keystone, top-order predator that are assumed to engage in cost-efficient movement and foraging patterns. To investigate the extent to which oscillatory diving by tiger sharks conform to these patterns, we used a biologging approach to model their cost of transport. High-resolution biologging tags with tri-axial sensors were deployed on 21 tiger sharks at Ningaloo Reef for durations of 5–48 h. Using overall dynamic body acceleration as a proxy for energy expenditure, we modelled the cost of transport of oscillatory movements of varying geometries in both horizontal and vertical planes for tiger sharks. The cost of horizontal transport was minimized by descending at the smallest possible angle and ascending at an angle of 5–14°, meaning that vertical oscillations conserved energy compared to swimming at a level depth. The reduction of vertical travel costs occurred at steeper angles. The absolute dive angles of tiger sharks increased between inshore and offshore zones, presumably to reduce the cost of transport while continuously hunting for prey in both benthic and surface habitats. Oscillatory movements of tiger sharks conform to strategies of cost-efficient foraging, and shallow inshore habitats appear to be an important habitat for both hunting prey and conserving energy while travelling.

# 1. Introduction

The tiger shark is a top-order predator that has wide-ranging impacts on the ecology of the tropical and warm-temperate marine ecosystems it inhabits [1,2]. These sharks are generalist feeders with anatomical specializations for feeding on large prey [3–7]. Although they actively hunt prey, they are also facultative scavengers [7], which avoids or reduces many of the energetic costs involved in the process of active predation [7,8]. However, injured or dead prey are windfalls that only occur sporadically and for the most part, tiger sharks must actively locate, hunt and capture their food. As large, top-order predators, they are assumed to do so in a cost-efficient manner [9,10].

Tagging studies have revealed that the fine-scale movements of tiger sharks are characterized by patterns of oscillatory ascents and descents (dives) throughout the water column [2,11], a behaviour that is shared by many species of large epipelagic fishes [12]. For tiger sharks, these dives have been recorded in a number of habitats, ranging from very shallow (less than 4 m) seagrass beds in Shark Bay [2], to deeper (greater than 100 m), offshore waters in Hawaii and Brazil [11,13,14]. It has been argued that these oscillatory movements are not likely to be primarily driven by behavioural thermoregulation, because they often occur in well-mixed and/or shallow environments [2,11] where there is little gradient in temperature between the surface and the seafloor. Instead, previous studies have suggested that oscillatory diving provides an effective search strategy for an animal that feeds on prey both at the surface and near the seabed [2,11]. More recently, tiger sharks tagged at Ningaloo Reef have been found to make oscillatory dives during the tortuous movements associated with area-restricted search [15], a search pattern that has been linked with increased foraging success in many marine animals [16–18]. Oscillatory dives by tiger sharks, however, also occurred throughout directional swimming, suggesting that factors other than foraging also contribute to these patterns [15].

In addition to allowing sharks to search for prey, oscillatory dives may represent a strategy to reduce the cost of transport. Weihs [19] proposed that a two-stage mode of swimming may allow negatively buoyant animals to reduce energy expenditure relative to swimming at a level depth between two horizontal points, whereby individuals oscillate throughout the water column, gliding on descent and actively swimming on ascent. Recent advances in the use of tri-axial sensors in biologging tags provide a means to test this hypothesis in a natural setting. Tri-axial accelerometers can be used to calculate dive angles in addition to overall dynamic body acceleration (ODBA), a parameter that can act as a proxy for energy consumption [20], and can, therefore, be used to quantify and compare the energy costs of different movement behaviours. Gleiss *et al.* [21] used a biologging approach to explore how differences in the dive geometry in whale sharks, *Rhincodon typus*, influenced the cost of transport with respect to both horizontal and vertical distance, using dynamic body acceleration as a proxy for power. Empirical optimality models suggested that some dive profiles reduced the cost of horizontal transport, while other steeper-angled dives minimized the cost of vertical transport; indicating that the choice of dive undertaken by an individual may be dependent on the ecological context, i.e. travel, search or foraging [22].

As tiger sharks are negatively buoyant [11], we hypothesize that they may also exploit their weight in water and use oscillatory movements to reduce the cost of transport while travelling or searching for prey. However, given that most dead prey will sink to the bottom, whereas active prey may be more easily ambushed at the surface where it can be approached from below [23], movement strategies may be habitat dependent. Tiger sharks tagged at Ningaloo Reef were found to undertake oscillatory movements in both inshore and offshore habitats [15]. In very shallow habitats (less than 3 m), tiger sharks would have no need to undergo vertical search patterns given the short distance between the surface of the water and the seabed. In deeper habitats, however, individuals would need to actively transit through a larger water column between the surface and seabed in their search for prey. Therefore, we predict that the diving energetics and kinematics of these animals will be dependent on the depth of the habitat being occupied, with deeper waters featuring steeper dive angles when searching for prey to allow for faster transit times between the surface and seabed.

Here, we use a biologging approach to examine the energetics of oscillatory diving behaviour in tiger sharks at Ningaloo Reef, Western Australia. We examine the incidence of gliding behaviour in tiger sharks and model the cost of transport of oscillatory movements of varying geometries using ODBA as a proxy for energy expenditure. We document the range of dive angles used by tagged sharks in relation to habitat depth, discuss the role of oscillatory movements in both efficient foraging and energy conservation, and re-evaluate the drivers of oscillatory diving in this species.

# 2. Material and methods

## 2.1. Data collection

Tiger sharks ($n = 22$) were captured and tagged at Ningaloo Reef, Western Australia in April and May 2017 following the methods described in Andrzejaczek et al. [15]. In brief, tiger sharks were captured using baited drumlines and secured alongside a 5.8 m vessel with the leader and tailrope. Either a CATS (Customized Animal Tracking Solutions, Australia) Diary Tag (dimensions and weight with clamp: $15 \times 4 \times 6$ cm and 300 g) or CATS Cam Tag ($23 \times 4 \times 7$ cm and 500 g) were then clamped to the dorsal fins for periods of 7–48 h (see electronic supplementary material, table S1). All tags were equipped with tri-axial accelerometers, magnetometers and gyroscopes, and sensors for depth, temperature and light. All sensors recorded continuously at 20 Hz. In addition, 14 of the 22 deployments recorded video at pre-programmed hours of the day for a maximum of 6 h per deployment. The tags detached from the clamp in the days following tagging, and were recovered using a handheld VHF receiver operated from a vessel.

## 2.2. Data processing

Data recorded by the tags were processed to obtain a number of parameters classifying shark movements, behaviour and the external environment. Vertical phases of movement, shark body pitch angles (orientation of the shark with regard to the horizontal plane), ODBA and tailbeat kinematics were calculated as described in Andrzejaczek et al. [15] (see electronic supplementary material for more detail). In brief, the depth record was split into vertical phases of ascent, descent and constant depth (more detail below). Tri-axial sensor data were then used to calculate pitch, ODBA, and the signal amplitude and frequency of tailbeats in Igor Pro v. 7.0.4.1 (Wavemetrics, Inc. Lake Oswego, USA). Tailbeat data were used to calculate the recovery period from capture following methods outlined by Whitney et al. [24]. Video data were also processed as per Andrzejaczek et al. [15], and were used to record interactions with prey, determine habitat types, and validate behaviours recorded by the sensors (e.g. tailbeats and gliding behaviour). In addition, sensor data were used to compute gliding behaviour and ascent and descent speed, as described below.

### 2.2.1. Depth record

Vertical velocity (VV), defined as the rate of change in depth over a 1 s period, was used to split the depth record into vertical swimming phases (ascending, descending and level swimming). This was executed by smoothing the depth record using a 10 s running mean and calculating the average VV by taking the difference of this smoothed depth between successive points at 1 s intervals. Ascents and descents were defined where VV exceeded an absolute value of $0.05$ m s$^{-1}$ for more than 10 s, and level where this value was not exceeded [15,24]. As the error in the depth sensor was minimal (less than ±10 cm), we do not believe sensor accuracy significantly affected vertical movement phase classification.

### 2.2.2. Gliding behaviour

We used a continuous wavelet transformation on the dynamic component of the sway (i.e. lateral) axis to calculate the signal amplitude and frequency of shark tailbeats using the angular velocity data [15,25]. These data were used to quantify the incidence of gliding behaviour—defined here as a cessation of tailbeats for more than 1 s—through a two-step process as per Andrzejaczek et al. [26]. Briefly, (i) gliding behaviour was isolated for each individual shark using the 'k-means cluster' function in the Ethographer for Igor Pro [25]. This function clustered the spectra computed by the wavelet transformation based on the similarity of shape. The behavioural spectrum with the lowest peaks in angular velocity signal amplitude was assumed to represent gliding behaviour [11], and the incidence of the resulting cluster was then inspected against the dynamic sway data. As this cluster did not match with gliding behaviour in some individuals (i.e. tailbeats evident in sway data were classified to be gliding, and vice versa), (ii) threshold values of angular velocity signal amplitude and tailbeat frequency were set using the characteristics of correctly classified gliding behaviour for each individual (from visual inspection of the dynamic sway data and concurrent videos). These thresholds were then used to extract glides from all sharks, and an additional manual quality control was added

in the rare case where the threshold obviously misclassified glides (electronic supplementary material, figure S1).

### 2.2.3. Ascent and descent speeds

Vertical velocity (VV) and pitch ($\varphi$) were used to estimate the mean speed of ascents and descents through trigonometry as per

$$\text{speed (m s}^{-1}) = \frac{\text{vertical velocity (m s}^{-1})}{\sin(\varphi)}. \tag{2.1}$$

This, however, could only be calculated when pitch exceeded 20° due to the large errors associated with estimating speed at low pitch angles [21].

### 2.2.4. Window size and statistics

The sampling window used for analysis was determined by calculating the time period where the highest variance in turning angles was observed, while being of sufficient size to capture the longest recorded dives in their entirety at all depths as per Andrzejaczek *et al.* [15] (see electronic supplementary material, figures S2, S3, and supplementary methods for more detail). This time window was estimated to be 15 min (900 s). Therefore, a number of vertical movement parameters were summarized for each 15 min window of each deployment including mean (±s.d.) and maximum depth, ascent pitch, descent pitch, ascent VV and descent VV. The per cent of time spent moving vertically (ascending and descending), termed the 'diving ratio', was also calculated within each window for each individual as per

$$\text{diving ratio} = \frac{\text{time vertically moving in window (s)}}{\text{total time in sampling window (900 s)}}. \tag{2.2}$$

## 2.3. Data analysis

### 2.3.1. Generalized linear mixed models

Generalized linear mixed models (GLMMs) were fitted using the `nlme` package in R v. 3.4.0 [27,28] to investigate possible relationships between seabed depth and vertical movement behaviours in tiger sharks. The use of mixed models allowed us to set individual tiger shark as a random intercept effect in all models and therefore control problems associated with non-independence due to repeat measurements of behaviour from the same individual [29]. The maximum depth ($m$) recorded within each time window was used as a proxy for seabed depth (based on video analysis; see Andrzejaczek *et al.* [15]) and was set as the only fixed explanatory variable for all models. Ascent pitch, descent pitch, ascent VV, descent VV and diving ratio were all set sequentially as response variables. As time-series data were inherently auto-correlated, we modelled the serial dependence in our data using an auto-regressive process of order 1 (AR1), which assumes that the magnitude of the data at time $t$ is affected by the magnitude of the data at time $t-1$ [22,30]. Auto-correlation was tested on the initial model fits without a correlation structure, revealing a steady decline of serial correlation with increasing lag from time $t$. The correlation at lag = 1 was then used in specifying the correlation structure of the data [30] and added as a final term to each model using the corAR1 function in R. Together with nautical charts from Ningaloo Reef, maximum depth was used to classify 15 min sampling windows as either 'inshore' (less than 25 m, inside the reef) or 'offshore' (greater than 25 m, outside the reef). Initial model runs considered the entire dataset; however, only four tiger sharks entered offshore habitats resulting in an unbalanced model design and therefore biasing predictions in deeper water to these individuals. In addition, residuals displayed unequal variance with depth (i.e. were heterogeneous). To overcome these issues, inshore and offshore periods were considered in separate GLMMs. The resulting models were compared against the null models and ranked using Akaike's information criterion (AIC; table 1), with the model with the lowest AIC subsequently selected. In addition, likelihood ratio tests were performed using the anova command between the nested models.

### 2.3.2. Cost of transport models

We modelled the cost of transport of oscillatory movements of varying geometries in relation to optimization of either horizontal or vertical distance travelled following methods similar to

**Table 1.** Results of generalized linear mixed models testing the relationship between seabed depth (MaxD) and vertical movement behaviours. All models were compared with null models using Akaike's information criterion (AIC) and conditional ($R^2c$) and marginal ($R^2m$) $R^2$ values. All models were run using the nlme package in R with shark identity included as a random effect. All null models included the random effect. Inshore indicates windows where the maximum depth was less than 25 m, and offshore where the maximum depth was greater than 25 m. p-Values indicate those obtained from likelihood ratio tests comparing nested models. Bolded models indicate those chosen in the model selection process (i.e. those with the lowest AIC).

| | d.f. | AIC | $R^2m$ | $R^2c$ | p-value |
|---|---|---|---|---|---|
| **inshore model** | | | | | |
| **diving ratio ∼ maximum depth** | 629 | −831 | 0.35 | 0.55 | <0.001 |
| diving ratio ∼ 1 | 629 | −629 | 0 | 0.42 | |
| **descent pitch ∼ maximum depth** | 629 | 2724 | 0.35 | 0.48 | <0.001 |
| descent pitch ∼ 1 | 629 | 2903 | 0 | 0.31 | |
| **ascent pitch ∼ maximum depth** | 629 | 2672 | 0.18 | 0.24 | <0.001 |
| ascent pitch ∼ 1 | 629 | 2743 | 0 | 0.14 | |
| **descent vertical velocity ∼ maximum depth** | 629 | −2695 | 0.26 | 0.53 | <0.001 |
| descent vertical velocity ∼ 1 | 629 | −2555 | 0 | 0.30 | |
| **ascent vertical velocity ∼ maximum depth** | 629 | −2941 | 0.2 | 0.36 | <0.001 |
| ascent vertical velocity ∼ 1 | 629 | −2845 | 0 | 0.18 | |
| **offshore model** | | | | | |
| diving ratio ∼ maximum depth | 173 | −165 | 0.09 | 0.39 | 0.38 |
| **diving ratio ∼ 1** | 173 | −168 | 0 | 0.37 | |
| descent pitch ∼ maximum depth | 173 | 1062 | 0.002 | 0.26 | 0.06 |
| **descent pitch ∼ 1** | 173 | 1054 | 0 | 0.27 | |
| ascent pitch ∼ maximum depth | 173 | 919 | 0.007 | 0.15 | 0.47 |
| **ascent pitch ∼ 1** | 173 | 912 | 0 | 0.11 | |
| descent vertical velocity ∼ maximum depth | 173 | −348 | 0.03 | 0.26 | 0.09 |
| **descent vertical velocity ∼ 1** | 173 | −360 | 0 | 0.35 | |
| ascent vertical velocity ∼ maximum depth | 173 | −466 | 0.004 | 0.21 | 0.48 |
| **ascent vertical velocity ∼ 1** | 173 | −482 | 0 | 0.23 | |

those described by Gleiss *et al.* [21] (figure 1). First, we calculated the total mechanical cost (TC) of an oscillation (an ascent (a) and descent (d) combined) in units of ODBA (g) using the equation

$$TC = T_a \times ODBA_a + T_d \times ODBA_d + k \times (T_d + T_a), \tag{2.3}$$

where $T_a$ and $T_d$ are the time spent ascending and descending, respectively, $ODBA_a$ and $ODBA_d$ are the ODBA of ascents and descents, respectively, and $k$ is a proxy for basal metabolic cost. Previous studies have shown that basal metabolic costs are approximately 60% of routine metabolic rate in sharks (see [31] and references therein), and therefore we estimated $k$ at 60% of the mean ODBA recorded for all sharks (0.026 g) or $k = 0.0156$ g. $T_a$, $T_d$, $ODBA_a$ and $ODBA_d$, were all calculated depending on pitch angle ($\varphi$), with ODBA estimates excluding those taken from the top 2 m due to the effects of wave action at the surface frequently creating superficially high ODBA levels here (confirmed by video data) [32]. $ODBA_a$ was estimated from the quadratic relationship between ODBA and $\varphi_a$, as a quadratic model was found to fit the data better than the equivalent linear model through model testing (figure 2a; $ODBA = 2 \times 10^{-5}\varphi_a^2 + 0.00001\varphi_a + 0.0226$). This data-driven approach was selected as we did not have the data to use a mechanistic approach to test how the efficiency of lift production changed as body pitch changed. For $ODBA_d$, a single mean value of ODBA during descents (0.012g) was used, as no relationship was found between $\varphi_d$ and ODBA

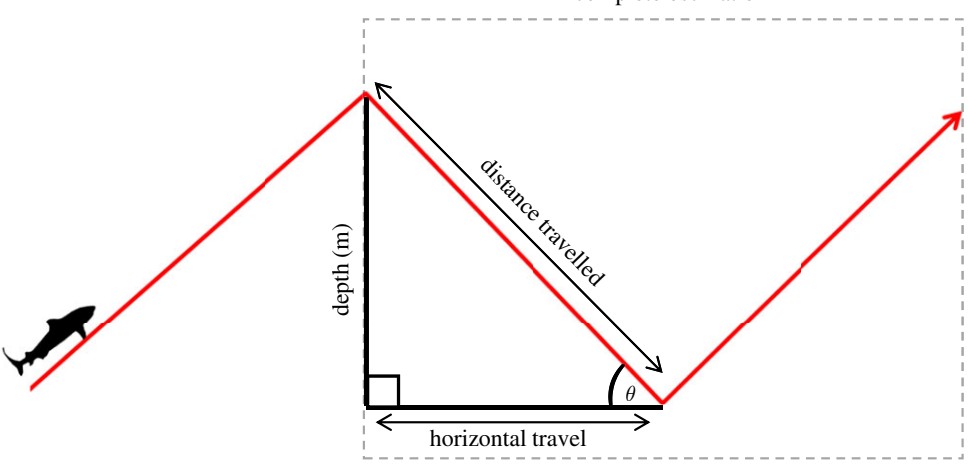

**Figure 1.** Schematic of an oscillatory movement, and the distance parameters used in calculating the cost of transport.

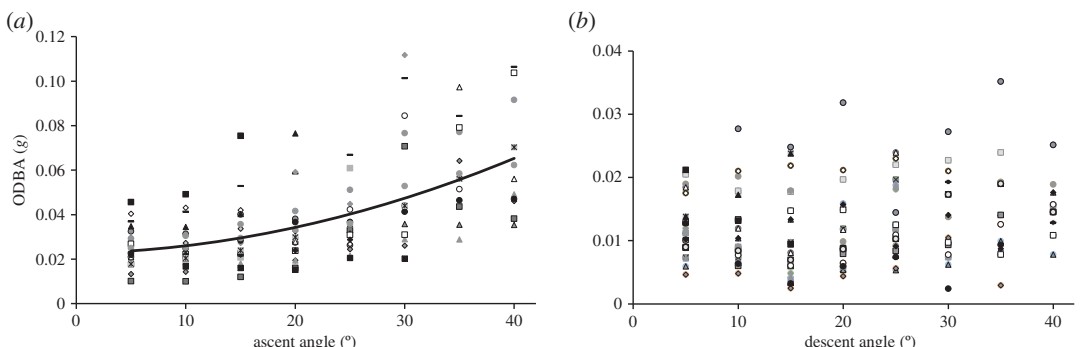

**Figure 2.** Mean instantaneous ODBA for all 21 tiger sharks (individuals represented by different symbols) at a range of absolute (*a*) ascent and (*b*) descent angles (placed in 5° bins). All sharks displayed a quadratic relationship with a positive slope for ODBA ~ ascent angle (ODBA $= 2 \times 10^{-05} \varphi_a^2 + 0.00001 \varphi_a + 0.0226$). No consistent relationship was found between ODBA and descent angle.

(figure 2*b*). $T_a$ and $T_d$ are a function of $\varphi_a$ and $\varphi_d$, respectively, depth and mean speed, and were calculated using the following equations:

$$T_a = \frac{\text{depth}/\sin(\varphi_a)}{\text{mean ascent speed}} \tag{2.4}$$

and

$$T_d = \frac{\text{depth}/\sin(\varphi_d)}{\text{mean descent speed}}. \tag{2.5}$$

We regressed swimming speed and body pitch where pitch was greater than 20° for each individual shark, and did not detect quantifiable changes in speed with pitch angle (electronic supplementary material, table S2). Where the relationship was found to be significant, slope did not exceed 0.02, and we, therefore, concluded that increased locomotory activity at increasing pitch was due to counteracting gravity, rather than an inherent increase in speed with pitch angle [21]. As a result, we used a fixed mean estimate of speed for both ascents (0.87 m s$^{-1}$) and descents (0.85 m s$^{-1}$). TC was calculated for fixed ascent angles from 5° to 45° at 5° increments. For each 5° increment of ascent angles, TC was calculated sequentially for descent angles of 5–20°, also at 5° increments.

We constructed two different models describing the cost of horizontal transport (COT$_{HD}$) and cost of vertical transport (COT$_{VD}$) for tiger sharks. These models calculated the cost of moving a unit of horizontal (HD) and vertical distance (VD), respectively, and were used to determine the angles that optimized the efficiency of transport on each of these scales. The COT$_{HD}$ was modelled by

$$\text{COT}_{HD} = \frac{\text{TC}}{\text{HD}}. \tag{2.6}$$

Where horizontal distance was calculated from ascent and descent pitch using the equation

$$HD = \frac{depth}{\tan(\varphi_d)} + \frac{depth}{\tan(\varphi_a)}. \tag{2.7}$$

The $COT_{VD}$ was modelled by

$$COT_{VD} = \frac{TC}{2 \times depth}. \tag{2.8}$$

All model calculations used oscillations of 10 m depth; however, the resulting COT for horizontal and vertical distance was the same regardless of depth.

# 3. Results

## 3.1. Track summary

Tag data were recovered from a total of 21 tiger sharks (ranging 2.65–3.8 m total length) at Ningaloo Reef (see electronic supplementary material, table S1 for full summary of tag deployments). One shark was recaptured after 11 days and was re-tagged (TS17 and TS24 in electronic supplementary material, table S1). We assumed a recovery period from capture and tagging by the sharks of 4 h based on an analysis of tailbeats, and for this reason, the first 4 h of each dataset were excluded from further analysis [15]. Evidence for recovery after 4 h was also provided by the video records, which showed investigation of prey and consumption of a discarded fish head by sharks within 2 h of release after tagging.

Tagged tiger sharks swam at a mean (±s.d.) depth of $10.46 \pm 10.21$ m, predominately residing in inshore habitats, with four tiger sharks moving into offshore habitats and one individual diving to a maximum depth of 94 m. Sharks moved vertically for a mean of $38.4 \pm 26\%$ of their track and used significantly steeper descent ($-11.11 \pm 3.6°$) than ascent ($9.38 \pm 2.6°$) angles (Wilcoxon signed-ranks test, $p < 0.001$).

## 3.2. Gliding behaviour

There was a high level of individual variation in both the proportion of gliding descents by tiger sharks (range 0–35%, figure 3) and the angle of descent while gliding. Six sharks glided for more than 19% of their total descent time, of which three glided for more than 30% of their descents. A maximum uninterrupted glide time of 2 min was recorded for one individual. Gliding was rarely recorded in the first 4 h of each tag deployment, (subsequently designated as the recovery period), and no gliding behaviour was recorded for two sharks (figure 3). In addition, only very short, unsustained (less than 3 s), periods of gliding were recorded on ascent, with 50% of sharks recording no gliding on ascent. Gliding on descent was exhibited by tiger sharks in both inshore and offshore zones and occurred at a mean angle of $-14.9 \pm 5.7°$, though this varied substantially among individuals (electronic supplementary material, figure S4). Mean descent angles were significantly steeper when individuals were gliding (paired Wilcoxon signed-rank test, $p < 0.001$), increasing the average angle of descent by $5.7 \pm 3.8°$.

## 3.3. Relationship between seabed depth and vertical movement behaviours

GLMMs revealed statistically significant relationships between seabed depth and vertical movements of tiger sharks in inshore habitats (table 1). For inshore GLMMs, all models retained their predictor variables of vertical movement (i.e. diving ratio, descent pitch, ascent pitch, descent VV and ascent VV), whereas for offshore models, only the random effect of tiger shark identity was retained. Up to depths of 25 m, diving ratio, pitch and VV all increased with seabed depth, after which point relationships plateaued and displayed high levels of variation (electronic supplementary material, figure S5). Inshore, maximum seabed depth explained 35% (marginal $R^2$) of the variation in both diving ratio and descent pitch. For all models, high conditional $R^2$ values (up to 55%) suggested high inter-individual variability in vertical movements (table 1). Mean absolute pitch angles increased between inshore and offshore habitats from $10.3 \pm 2.8°$ to $14.1 \pm 4.7°$ for descent, and $9 \pm 2.3$ to $10.7 \pm 3.2°$ for ascent (figure 4a,b). The mean diving ratio increased from $31 \pm 23\%$ to $69 \pm 17\%$ between

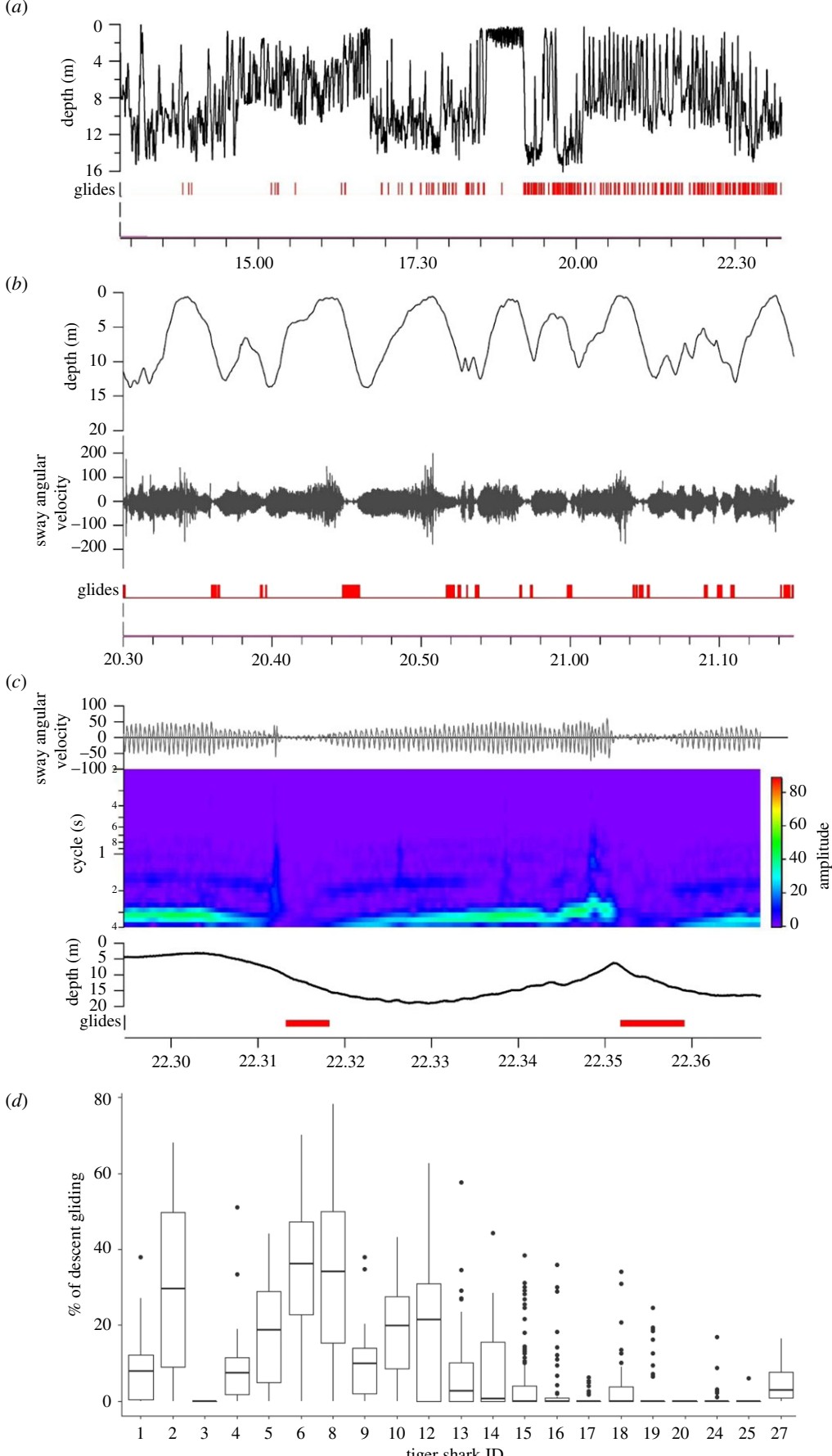

**Figure 3.** (*Caption opposite.*)

**Figure 3.** (*Opposite.*) Gliding in tiger sharks over the course of (*a*) an entire track (10 h); (*b*) 45 min; and (*c*) 7 min. Glides are marked in red on the *x*-axis. Dynamic angular velocity was taken from the lateral (sway) axes recorded by the gyroscope. Cycle on the *y*-axis of (*c*) represents the inverse of tailbeat frequency (i.e. the length of time it takes to complete one tailbeat). (*d*) The proportion of descent time spent gliding in a 15 min window by each individual tiger shark.

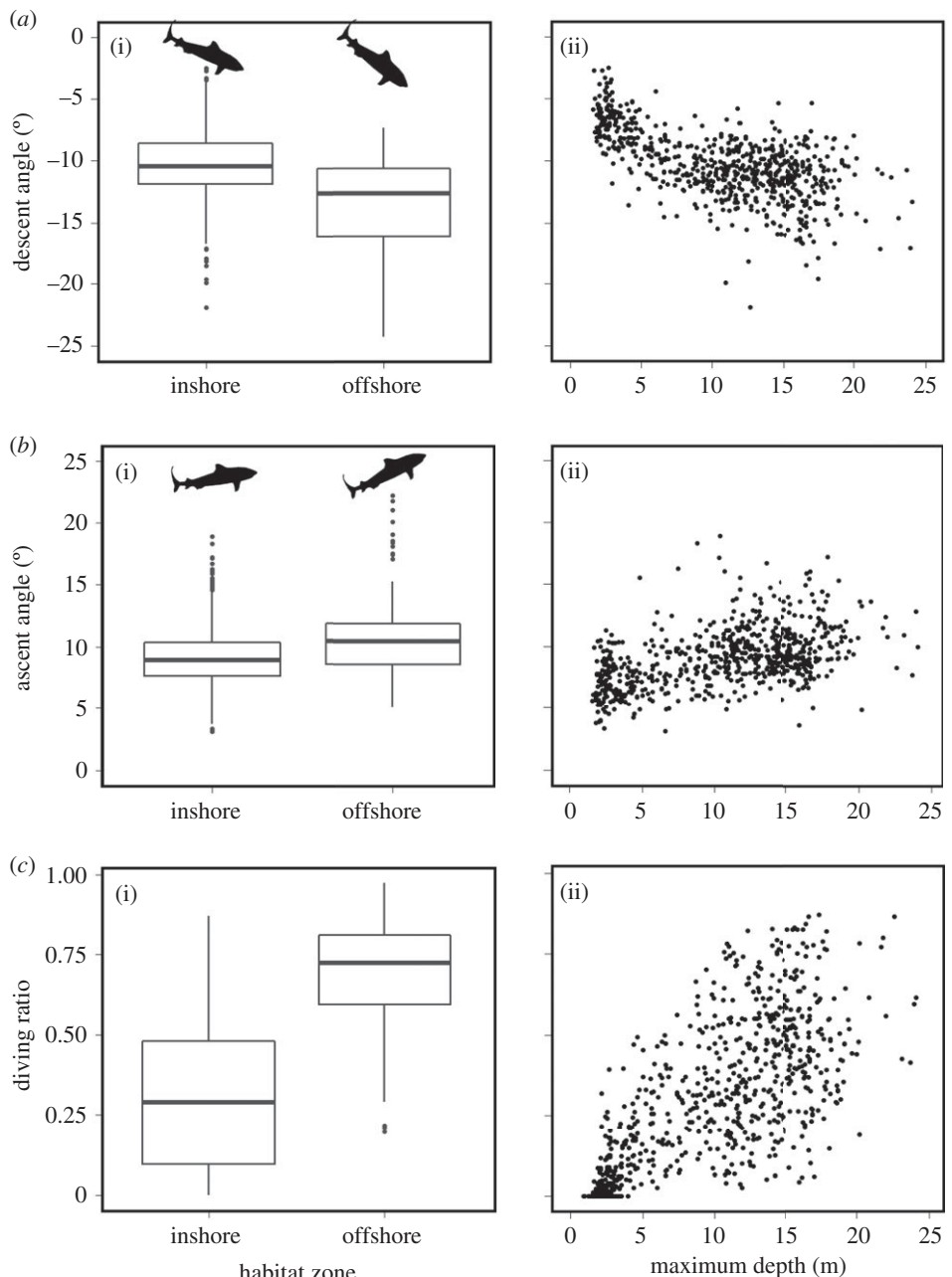

**Figure 4.** Relationship between vertical movement behaviours and habitat depth. Boxplots in (i) display differences in (*a*) descent angle, (*b*) ascent angle and (*c*) diving ratio between habitat zones. Inshore denotes areas where maximum depth within a window was less than 25 m, and offshore where depths were greater than 25 m. Shark silhouette angles are not to scale in (*a*) and (*b*). Scatterplots in (ii) display relationships between maximum depth in a sampling window, and (*a*) descent angle, (*b*) ascent angle and (*c*) diving ratio, in inshore zones (less than 25 m).

inshore and offshore habitats (figure 4*c*). The diving ratio increased quickly between 5 and 20 m depths, after which the slope rapidly decreased with increasing depth.

## 3.4. Cost of transport models

The mechanical cost of ascending was greater than that for descending (one-way ANOVA, $F_{1,240002} = 1635$, $p < 0.0001$). The cost of level swimming by tiger sharks could not be calculated as the effects of

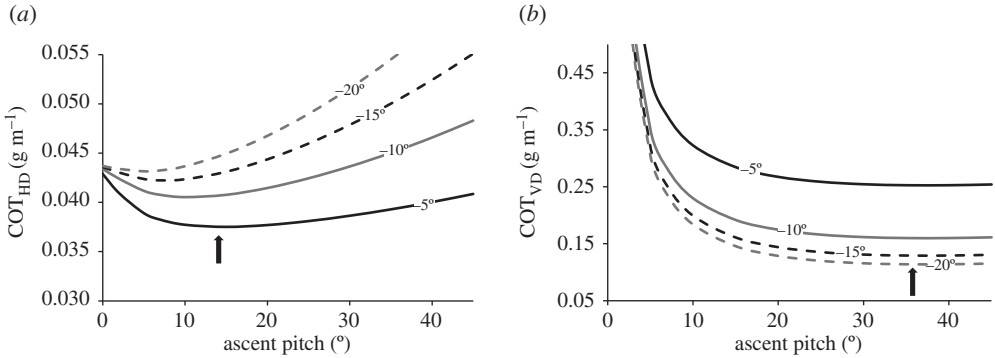

**Figure 5.** The horizontal cost of transport (*a*) and the vertical cost of transport (*b*) for a tiger shark executing a single oscillatory movement from the surface to 10 m according to descent and ascent pitch. Lines represent the cost of a given descent pitch angle (label on figure) in relation to varying ascent pitch (*x*-axis). Arrows indicate where the cost of transport is minimized in both cases. Reduction of the cost of transport in the horizontal (COT$_{HD}$) favours descending at minimal pitch during descent and ascending at an angle of 5–14°. Reduction of vertical travel costs (COT$_{VD}$) occurred by ascending and descending at steeper angles, although potential energy savings at the absolute ascent and descent angles steeper than 20° and 15°, respectively, are minimal.

wave action at the surface frequently created superficially high ODBA levels that could not be removed reliably from the data. These effects of surface waves were confirmed by the video data.

For all sharks, a positive quadratic relationship was found between mean binned ascent angle and ODBA (figure 2*a*, $R^2 = 0.96$). This relationship had positive slopes for all individual sharks with a mean $R^2$ of $0.8 \pm 0.23$ (range 0.12–0.99). No consistent relationship was found between mean binned descent angle and ODBA (figure 2*b*). We regressed swimming speed and body pitch where pitch was greater than 20° for each individual shark, and did not detect quantifiable changes in speed with pitch angle (electronic supplementary material, table S2).

We used the relationship between ascent pitch angle and ODBA to calculate ODBA as a proxy for mechanical cost. This was done at 5° intervals for COT models, and a single mean value of ODBA was used for descents. As gliding behaviour was highly variable and we found no evidence of accelerating VV towards the maximum depth of dives, we did not calculate the terminal velocity as per Gleiss *et al.* [21]. The total cost of an oscillation of given ascent and descent angles was then standardized by the corresponding vertical and horizontal distances travelled in order to calculate the cost of transport on each respective plane. We found that the cost of transport was reduced on both horizontal and vertical planes at absolute ascent and descent angles greater than 0° (figure 5*a*), meaning that vertical oscillations conserved energy compared to swimming at a level depth. The cost of transport for tiger sharks in the horizontal plane was minimized by descending at the lowest possible angle, and ascending at an angle of 5–14°. The cost of transport in the vertical plane was optimized at steeper ascent angles of 35°, and decreased with steeper descent angles; however, the potential energy savings at absolute ascent and descent angles steeper than 20° and 15°, respectively, were minimal (figure 5*b*).

# 4. Discussion

Our study provides evidence that oscillatory movements through the water column allowed tiger sharks to conserve energy while travelling and searching for prey and result in a strategy of cost-efficient foraging. Our empirical models of the cost of transport showed that shallow angles of ascent and descent allowed tiger sharks to reduce their cost of transport in the horizontal plane relative to swimming at a level depth, whereas steeper angles reduced their energetic costs in the vertical plane. The absolute dive angles of tiger sharks increased between inshore and offshore zones, presumably to reduce energy costs while continuously hunting for prey in both benthic and surface habitats in a coastal environment.

## 4.1. Cost of vertical movements

Patterns of gliding varied among tagged sharks. Gliding occurred through 0–35% of the overall descent phase of oscillations for individuals at Ningaloo, a finding comparable to that of Nakamura *et al.* [11]. Although the majority of sharks displayed increasing levels of gliding behaviour post-release, as previously observed in blacktip sharks [24], some sharks—including an individual tagged for more than 15 h—recorded no gliding behaviour throughout tag deployment. In addition, angles at which

gliding behaviour was recorded differed substantially among tagged sharks. This variability may be driven by minor individual differences in body composition [33–35], and/or behavioural differences in their response to catch and release. At all spatial scales, movements of tiger sharks have been characterized by high levels of individual variation [36,37].

The energetic costs involved in descents were significantly lower than those of ascents, despite the majority of the descent phases of oscillations involving active swimming. This reflects the negative buoyancy of these animals [11], as this enables their descent phase to be partially powered by gravity. When ascents and descents were considered together and compared to swimming an equivalent horizontal distance at a level depth, the modelled cost of horizontal transport was minimized through the former, two-stage mode of oscillatory swimming. For tiger sharks, this was minimized at a descent angle of −5° and ascent angle of 14°, values that closely match modelled dive angles of whale sharks, where horizontal transport was optimized at −10° and 11° for descents and ascents, respectively [21]. For whale sharks, costs were minimized at the smallest angle at which whale sharks could glide. The mean descent angle of tiger sharks was significantly steeper when tagged individuals exhibited passive gliding descents, and we suspect that steeper optimal descent angles may exist for individuals that display a greater degree of gliding behaviour. Reduction of vertical travel costs for tagged tiger sharks also exhibited a similar pattern to whale sharks, whereby steeper angles on ascent than descent reduced the cost of vertical transport. Ascent angles of 35° were found to reduce the cost of vertical transport for tiger sharks at Ningaloo, a marked increase from the 23° reported for whale sharks [21], although energy savings beyond 20° appear to be minimal for tiger sharks. The costs of vertical transport decreased with increasingly steep descent angles, but again, energy savings at angles beyond −15° appeared to be minimal.

## 4.2. Cost-efficient foraging

Oscillatory movements of tiger sharks have previously been thought to be a strategy to allow them to alternate between searching the surface and seabed for prey [2,11]. Heithaus *et al.* [2] found that tiger sharks often spend a short period of time swimming at the surface before descending to swim along the seabed. After another short period they ascend, presumably searching for prey at the surface during the time they reside there. These patterns were consistent with those observed in our study. The diving ratio of tiger sharks tagged at Ningaloo Reef increased as seabed depth increased, i.e. sharks spent more time within a 15 min period ascending and descending relative to level swimming when they were in deeper water. Larger distances need to be traversed between the surface and seabed in deeper water, meaning a greater proportion of time is spent moving vertically in these areas. It is thus more likely that tiger sharks were in fact spending a similar length of time searching the surface and benthos for prey in between oscillations, regardless of water depth.

In the shallowest, inshore waters (less than 3 m), tiger sharks would be able to continuously hunt as either the seabed or the surface would always remain within their field of view. However, as they move into deeper water, these animals would need to transit through a larger water column between surface and benthic habitats in their hunt for prey or to scavenge, reducing the amount of time spent foraging and increasing the energy costs of this activity. Here, we found that the dive angles used by tiger sharks at Ningaloo increased as these animals moved into deeper waters, perhaps allowing them to compensate for the increased cost of hunting through a larger water column and reduce the amount of time spent transiting through open water. In shallower waters, tiger sharks predominately used shallow dive angles, consistent with movements that reduced the horizontal cost of transport. We cannot, however, discount the possibility that there are other mechanisms driving the differences in diving angles between habitats. For example, ascent angles were shallower than descent angles and remained relatively constrained between habitats, which may be a function of retinal specialization in the visual system of tiger sharks [38]. These sharks are able to observe their upper visual field in greater detail [38], and their angle of ascent may optimize the field of view for searching for unsuspecting prey silhouetted above them, whereas descent angles may be steeper to facilitate scanning the seabed below them. Nevertheless, this does not change the fact that, from a cost of transport standpoint, these animals were moving at cost-efficient angles in shallow habitats.

## 4.3. Model limitations

The cost of transport models constructed for tiger shark from acceleration data incorporate some parameters that do not allow for the accurate measurement of movement costs in this species.

Specifically, errors are likely to occur in the estimate of basal cost ($k$), as this has not been calculated for tiger sharks, and in the estimates for speed, as this was only calculated where absolute pitch angles exceeded 20°. However, Gleiss *et al.* [21] tested the sensitivity of the cost of transport models to predicting qualitative trends for whale sharks and found that (i) both doubling and halving of basal cost had no appreciable change on optimal ascent angle, and (ii) variations in the speed of 0.2 m s$^{-1}$ either side of the mean descent speed only changed optimal ascent angle by 2°. In addition, our speed estimates (0.85 m s$^{-1}$ for descent and 0.87 m s$^{-1}$ for ascent) fit into the range of those directly quantified by Nakamura *et al.* [11] (overall mean speed 0.54–0.92 m s$^{-1}$ among individuals, 0.53–0.82 m s$^{-1}$ when gliding). Therefore, we believe that the overall cost of our transport model for tiger sharks is relatively insensitive to inherent errors, and that model results are likely to accurately reflect the trends in dive pitch angles for optimizing the cost of vertical and horizontal transport.

## 4.4. Seascape effects on energy expenditure and gain

Our results suggest that very shallow waters (less than 3 m) are the most cost-efficient place for tiger sharks to travel and search for prey and may, therefore, represent an important energy landscape for these animals. Variation in an energy landscape occurs where the cost of transport is influenced by the physical characteristics of the environment [39]. This results in animals modifying movements among habitats to either exploit opportunities for energy gain, and/or reduce costs [40]. Establishing the variability in movement costs across landscapes can help us to understand how and why animals distribute themselves in space [39]. For tiger sharks, alternating between the seabed and surface in the search for prey means that the energetic costs of foraging increase with depth. Very shallow environments (less than 3 m), such as the sandflats of Ningaloo Reef, represent a landscape where tiger sharks can move and forage at the most cost-efficient dive angles. It is important to note, however, that energy inputs, or more specifically, food assimilation, also represent a key component of bioenergetics modelling and habitat profitability estimations [41]. Deeper environments may become more energetically profitable to sharks when prey densities are higher than in shallow environments and the energetic costs of moving sub-optimally are outweighed. Although the data are not available to model food availability in this system, data from pseudo-tracks, video and the ecotourism industry have highlighted the importance of the shallow sandflat environments at Ningaloo Reef for foraging tiger sharks [15]. In these habitats, tiger sharks often displayed the tortuous movement patterns thought to denote foraging behaviour [17], and predator–prey interactions have frequently been observed. Tiger sharks tagged in other regions have also been found to display a preference for shallow habitats, including seagrass habitats less than 4 m in depth in Shark Bay, and lagoon habitats in Hawaii [2,42].

The preferential selection of these shallow water habitats has implications for the distribution of predation pressure, and therefore the distribution of prey species. A study conducted over 15 years in the largely seagrass habitat of Shark Bay, Western Australia, found that the presence of tiger sharks influenced the movements and behaviour of a number of prey species, including turtles and dugongs [43], both of which are important grazers that exist in significant numbers at Shark Bay and Ningaloo Reef [44]. Given the prolonged residency of tiger sharks at Ningaloo Reef (several months; [37]) it seems likely that they will have similar structuring roles in this coral reef environment.

## 5. Conclusion

Collectively, our data showed that the oscillatory movements of coastal tiger sharks are the product of a combined strategy to search for prey while reducing the cost of transport. Tiger sharks modified their vertical movement behaviours between inshore and offshore zones, presumably to conserve energy while foraging for benthic and air-breathing prey. The shallow sandflats of Ningaloo Reef appear to be an important habitat for foraging and energy conservation.

Ethics. All methods used were in accordance with approved guidelines by the University of Western Australia Animal Ethics Committee (RA/3/100/1437), and under permit numbers 2881 (WA Department of Primary Industries and Regional Development) and 08-000322-3 (WA Department of Biodiversity, Conservation and Attractions).
Data accessibility. Raw data and GLMM code are accessible from the Dryad Digital Repository: https://doi.org/10.5061/dryad.vmcvdncqb [45].
Authors' contributions. S.A., M.G.M. and A.C.G. conceived the study. S.A., A.C.G., T.K.C. and K.O.L. performed fieldwork/data collection. S.A., A.C.G., C.P. and M.G.M. analysed and interpreted the data, and S.A. led the

writing of the manuscript. All authors contributed to drafting the article, and reading and approving the final manuscript.

Competing interests. The authors declare no competing or financial interests.

Funding. Fieldwork and tags were funded by crowdfunding on the Experiment platform (doi:10.18258/7190), a Holsworth Wildlife Research Endowment, a UWA Graduate Research School fieldwork award, BigWave Productions and the Australian Institute of Marine Science.

Acknowledgements. S.A. was supported by an Australian Government Research Training Program (RTP) Scholarship and an UWA top-up scholarship. We thank Murdoch University and Coral Bay Research Station for accommodation and vessel use while conducting fieldwork. Additional logistical support was provided by Olwyn Hunt and Louise Scott from the Australian Institute of Marine Science. This work would not have been possible without the generous support of numerous volunteers, particularly: Frazer McGregor, Abraham Sianipar, Adam Jolly, Blair Bentley, Evan Byrnes, Garry Teesdale and Michael Tropiano providing valuable field support; Olivia Seeger and Abraham Sianipar who helped with the video analysis. Nikolai Liebsch provided invaluable assistance with tag functioning. Thanks to Lauren Peel for helpful comments on an earlier version of this manuscript. We also thank two anonymous reviewers for providing constructive and insightful comments that improved the manuscript.

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
