## [Reviewer comments · Royal Society Open Science]

Review History

RSOS-191258.R0 (Original submission)

Review form: Reviewer 1

Is the manuscript scientifically sound in its present form?

Yes

Are the interpretations and conclusions justified by the results?

Yes

Is the language acceptable?

Yes

Do you have any ethical concerns with this paper?

No

Have you any concerns about statistical analyses in this paper?

No

Recommendation?

Accept with minor revision (please list in comments)

Comments to the Author(s)

This manuscript explained how tiger sharks change depth movement according to the depth of the sea. The model looks a little rough (The estimated optimal pitch angle does not appear to match the actual shark pitch angle), but I think it's worth publishing because the results are not so strange (I think it's interesting that the behavior changes depending on the habitat.). There were some wondering parts, but it was good because the explanation came out later.

Specific comments

L89 Although it is written to refer to Andrzejaczek et al. 2019, it is better to write the model and specifications (size, weight etc.) of the devices.

L98 Although it is written to refer to Andrzejaczek et al. 2019, the definition of vertical phase of movements (how to distinguish descent, ascent and horizontal swimming. From pitch? change of depth?) should be stated here.

L100 It is written as the pitch angle of vertical movement, but isn't it a shark angle? Is the shark moving parallel to the body axis (no angle of attack)?

L119 Is there a tailbeat frequency even during gliding? Is it okay to extract gliding with the same "mask" for all individuals, even though there are slight differences in the attaching position and differences in individual sizes?

L122 Although the ascent and descent speeds were calculated only when the pitch angle was 20° or more, the other analyses that appear later have a pitch angle smaller than 20° (e.g. Fig. 4). The validity when the pitch angle is shallower than 20° should be shown.

L128 Again, it is necessary to define vertical and horizontal swimming.

L172 It is necessary to state what formula you used to regress. Although it is written as a quadratic curve in the result, is it part of a sine curve due to the relationship between gravity and angle?

L179 No relationship between pitch angle and speed is declared here. Although it is written in the result, I think that it is better to quote the table S2 here and explain the reason.

L208 Are there any gliding ascent?

L232 Is it possible to calculate the cost of level swimming along the seabed?

L272 It is better to cite this paper because it is a shark paper rather than a seal one. Del Raye et al. (2013) Travelling light: white sharks (*Carcharodon carcharias*) rely on body lipid stores to power ocean-basin scale migration. *Proc R Soc B* 280:20130836. And this sentence has no verb.

L286 It is better to examine the relationship between glide rate and pitch angle.

L304 Isn't it strange to use the same vertical velocity when estimating the dive ratio in Fig. 5 despite the result that the pitch angle changes between inshore and offshore?

L314 Looking at Fig. 4, the pitch angle changes continuously as it gets deeper (especially ascent), so I think there should be a discussion about that. Is it possible to trade off the horizontal and vertical directions with a shallower pitch angle than the model that covers the vertical direction?

Fig. 1 How about overlaying glides on a depth profile (e.g. colouring only glides)?

Fig. 4 The shark silhouette is steep even though the offshore ascent angle is smaller. In results, "Mean absolute pitch angles increased between inshore and offshore habitats from 9 ± 2.3 to 10.7 ± 3.2 for ascent." Has inshore and offshore been switched?

Fig. S1 It is better to swap left and right so that it fits in Fig. 4.

Review form: Reviewer 2

Is the manuscript scientifically sound in its present form?

No

Are the interpretations and conclusions justified by the results?

No

Is the language acceptable?

Yes

Do you have any ethical concerns with this paper?

No

Have you any concerns about statistical analyses in this paper?

Yes

Recommendation?

Reject

Comments to the Author(s)

Dear Dr Andrzejaczek,

I have read your manuscript, which is on the whole well-written and raises an interesting topic of oscillatory movements in tiger sharks, provides measurements, and makes inferences about energy efficiency of movement during these "dives", with respect to dive angles.

There are some aspects of your work that cause me to have concerns about the validity of the conclusions drawn from the data you present here. Perhaps I haven't understood correctly, but you claim that tiger sharks adopt optimal dive angles during oscillatory movements. I am uncomfortable with this claim because optimality is a very strong word, meaning something quite specific, which I don't think you have measured. For example, you have not mentioned or measured the potential profitability of the different environments that sharks encountered. I'm not suggesting that you ought to have measured that; in general it's hard to do and most of us don't have that information about the environment our study species travel to. However, I think it's essential to mention and discuss it, and very clearly define in what sense you consider movements optimal. In addition, you make a comparison between inshore and offshore habitats, which is treated as a categorical variable. From figure 4, it appears that there was sufficient data to treat depth as a continuous variable, which would make better use of the resolution of the data and be more convincing.

The statistical modelling aspect of your work appears generally sound, but it's difficult to tell without more detail. You have not provided the code you used to carry out the modelling, so at the moment it's not reproducible even though you have provided the data. I would encourage you to always submit your code, in keeping with the ethos of reproducibility, without which a reader can't replicate your work. One question that lingers in my mind is why you have used a one-way ANOVA to compare the mechanical cost of ascending versus descending. Perhaps I've missed something, but the only reason to do a one-way ANOVA instead of the two-way ANOVA is if it was impossible to have an effect go in both directions (bigger and smaller). That doesn't seem to be the case here. Can you explain why you used a one-way ANOVA?

To summarise my main concerns with the paper:

1) the use of the word optimal is problematic, because it implies something very specific, and which I don't think you can back up, owing to the lack of information about how profitable a habitat is. My argument is that optimality implies a balance of input and output. In this manuscript you discuss measurements of output - energy spent - but you have no measurements of input - energy gained, which is the usual situation in studies of animal movement. My issue isn't that you lack these data, but rather that you don't discuss the issue of varying profitability. It may become worth spending extra energy and moving "sub-optimally" if the returns outweigh the costs. This isn't mentioned in your manuscript.

2) one of the central outcomes of this work appears to be that tiger sharks dive at different angles in shallow and deep water, shallow sandflats being mentioned as important habitat where sharks

can hunt while maintaining optimal angles. I find this conclusion problematic because most of the sharks in your study have length over 2.5m. This means that they physically cannot make descents or ascents past some maximum absolute angle. Perhaps I have misunderstood something here but again, the text needs to be made much clearer if that's the case.

3) the methodology and analysis are not well documented. As far as the data preparation is concerned, I would have liked to see much more detail, perhaps in an appendix, about how the data were processed and prepared for analysis, and details about the tag measurements. For example, what was considered a dive? What was the measurement error of the instrument? In addition, I would have liked to see a more comprehensive study of the effect of different time windows that could support the choice of time window in the analysis. Similarly, I would like to see a much more detailed rationale of the statistical modelling, including reasons for choosing the model you present, stating clearly the response variables and covariates, candidate models, error distributions and error structures, reason for choosing an autocorrelation function to model the errors and the way the final model was chosen, including a mention of the limitations of using AIC to do model selection. All of this could go in an appendix but it's critical that you explain it somewhere.

I have attached a document with comments on the manuscript (Appendix A).

Decision letter (RSOS-191258.R0)

08-Oct-2019

Dear Dr Andrzejaczek:

Manuscript ID RSOS-191258 entitled "Depth dependent dive kinematics suggest cost-efficient foraging strategies by tiger sharks" which you submitted to Royal Society Open Science, has been reviewed. The comments from reviewers are included at the bottom of this letter.

In view of the criticisms of the reviewers, the manuscript has been rejected in its current form. However, a new manuscript may be submitted which takes into consideration these comments.

Please note that resubmitting your manuscript does not guarantee eventual acceptance, and that your resubmission will be subject to peer review before a decision is made.

Your resubmitted manuscript should be submitted by 06-Apr-2020. If you are unable to submit by this date please contact the Editorial Office.

Best regards,
Lianne Parkhouse
Royal Society Open Science
openscience@royalsociety.org

on behalf of Professor Len Thomas (Associate Editor) and Professor Kevin Padian (Subject Editor)
 openscience@royalsociety.org

Associate Editor Comments to Author (Professor Len Thomas):

Dear Dr. Andrzejaczek, Thank-you for submitting your paper to RSOS. We received two reviews and, while the first reviewer raised only minor concerns, the second raised three major concerns, all of which I concur with. I am recommending rejection of your paper on this basis, but I invite you to re-submit if you can address all of these concerns, including adding (perhaps in an appendix) much more detailed information about your analytical methods. Please note that the second reviewer has also made many helpful comments on the manuscript itself. With best wishes, Len Thomas

Reviewers' Comments to Author:

Reviewer: 1

Comments to the Author(s)

This manuscript explained how tiger sharks change depth movement according to the depth of the sea. The model looks a little rough (The estimated optimal pitch angle does not appear to match the actual shark pitch angle), but I think it's worth publishing because the results are not so strange (I think it's interesting that the behavior changes depending on the habitat.). There were some wondering parts, but it was good because the explanation came out later.

Specific comments

L89 Although it is written to refer to Andrzejaczek et al. 2019, it is better to write the model and specifications (size, weight etc.) of the devices.

L98 Although it is written to refer to Andrzejaczek et al. 2019, the definition of vertical phase of movements (how to distinguish descent, ascent and horizontal swimming. From pitch? change of depth?) should be stated here.

L100 It is written as the pitch angle of vertical movement, but isn't it a shark angle? Is the shark moving parallel to the body axis (no angle of attack)?

L119 Is there a tailbeat frequency even during gliding? Is it okay to extract gliding with the same "mask" for all individuals, even though there are slight differences in the attaching position and differences in individual sizes?

L122 Although the ascent and descent speeds were calculated only when the pitch angle was 20° or more, the other analyses that appear later have a pitch angle smaller than 20° (e.g. Fig. 4). The validity when the pitch angle is shallower than 20° should be shown.

L128 Again, it is necessary to define vertical and horizontal swimming.

L172 It is necessary to state what formula you used to regress. Although it is written as a quadratic curve in the result, is it part of a sine curve due to the relationship between gravity and angle?

L179 No relationship between pitch angle and speed is declared here. Although it is written in the result, I think that it is better to quote the table S2 here and explain the reason.

L208 Are there any gliding ascent?

L232 Is it possible to calculate the cost of level swimming along the seabed?

L272 It is better to cite this paper because it is a shark paper rather than a seal one. Del Raye et al. (2013) Travelling light: white sharks (*Carcharodon carcharias*) rely on body lipid stores to power ocean-basin scale migration. *Proc R Soc B* 280:20130836. And this sentence has no verb.

L286 It is better to examine the relationship between glide rate and pitch angle.

L304 Isn't it strange to use the same vertical velocity when estimating the dive ratio in Fig. 5 despite the result that the pitch angle changes between inshore and offshore?

L314 Looking at Fig. 4, the pitch angle changes continuously as it gets deeper (especially ascent), so I think there should be a discussion about that. Is it possible to trade off the horizontal and vertical directions with a shallower pitch angle than the model that covers the vertical direction?
 Fig. 1 How about overlaying glides on a depth profile (e.g. colouring only glides)?
 Fig. 4 The shark silhouette is steep even though the offshore ascent angle is smaller. In results, "Mean absolute pitch angles increased between inshore and offshore habitats from 9 ± 2.3 to 10.7 ± 3.2 for ascent." Has inshore and offshore been switched?
 Fig. S1 It is better to swap left and right so that it fits in Fig. 4.

Reviewer: 2

Comments to the Author(s)

Dear Dr Andrzejczek,

I have read your manuscript, which is on the whole well-written and raises an interesting topic of oscillatory movements in tiger sharks, provides measurements, and makes inferences about energy efficiency of movement during these "dives", with respect to dive angles.

There are some aspects of your work that cause me to have concerns about the validity of the conclusions drawn from the data you present here. Perhaps I haven't understood correctly, but you claim that tiger sharks adopt optimal dive angles during oscillatory movements. I am uncomfortable with this claim because optimality is a very strong word, meaning something quite specific, which I don't think you have measured. For example, you have not mentioned or measured the potential profitability of the different environments that sharks encountered. I'm not suggesting that you ought to have measured that; in general it's hard to do and most of us don't have that information about the environment our study species travel to. However, I think it's essential to mention and discuss it, and very clearly define in what sense you consider movements optimal. In addition, you make a comparison between inshore and offshore habitats, which is treated as a categorical variable. From figure 4, it appears that there was sufficient data to treat depth as a continuous variable, which would make better use of the resolution of the data and be more convincing.

The statistical modelling aspect of your work appears generally sound, but it's difficult to tell without more detail. You have not provided the code you used to carry out the modelling, so at the moment it's not reproducible even though you have provided the data. I would encourage you to always submit your code, in keeping with the ethos of reproducibility, without which a reader can't replicate your work. One question that lingers in my mind is why you have used a one-way ANOVA to compare the mechanical cost of ascending versus descending. Perhaps I've missed something, but the only reason to do a one-way ANOVA instead of the two-way ANOVA is if it was impossible to have an effect go in both directions (bigger and smaller). That doesn't seem to be the case here. Can you explain why you used a one-way ANOVA?

To summarise my main concerns with the paper:

1) the use of the word optimal is problematic, because it implies something very specific, and which I don't think you can back up, owing to the lack of information about how profitable a habitat is. My argument is that optimality implies a balance of input and output. In this manuscript you discuss measurements of output - energy spent - but you have no measurements of input - energy gained, which is the usual situation in studies of animal movement. My issue isn't that you lack these data, but rather that you don't discuss the issue of varying profitability. It may become worth spending extra energy and moving "sub-optimally" if the returns outweigh the costs. This isn't mentioned in your manuscript.

2) one of the central outcomes of this work appears to be that tiger sharks dive at different angles in shallow and deep water, shallow sandflats being mentioned as important habitat where sharks

can hunt while maintaining optimal angles. I find this conclusion problematic because most of the sharks in your study have length over 2.5m. This means that they physically cannot make descents or ascents past some maximum absolute angle. Perhaps I have misunderstood something here but again, the text needs to be made much clearer if that's the case.

3) the methodology and analysis are not well documented. As far as the data preparation is concerned, I would have liked to see much more detail, perhaps in an appendix, about how the data were processed and prepared for analysis, and details about the tag measurements. For example, what was considered a dive? What was the measurement error of the instrument? In addition, I would have liked to see a more comprehensive study of the effect of different time windows that could support the choice of time window in the analysis. Similarly, I would like to see a much more detailed rationale of the statistical modelling, including reasons for choosing the model you present, stating clearly the response variables and covariates, candidate models, error distributions and error structures, reason for choosing an autocorrelation function to model the errors and the way the final model was chosen, including a mention of the limitations of using AIC to do model selection. All of this could go in an appendix but it's critical that you explain it somewhere.

I have attached a document with comments on the manuscript.

Author's Response to Decision Letter for (RSOS-191258.R0)

See Appendix B.

RSOS-200789.R0

Review form: Reviewer 1

Is the manuscript scientifically sound in its present form?

Yes

Are the interpretations and conclusions justified by the results?

Yes

Is the language acceptable?

Yes

Do you have any ethical concerns with this paper?

No

Have you any concerns about statistical analyses in this paper?

No

Recommendation?

Accept with minor revision (please list in comments)

Comments to the Author(s)

All peer review comments appeared to have been adequately addressed. I have only a few points to be made.

L309: I understand that there is a lowest angle for gliding, but I don't understand what the lowest possible angle is for tailbeating even during descent. The lowest descent angle where the ODBA is significantly smaller than the level swimming? There seems a unique descent angle for each shark in Fig.S4 and this represents to be the lowest possible angle? If there is a unique descent angle for each shark, the comparison to the model should be made to see if the ascent angle of each shark is consistent with the model that minimizes the COTHD for each shark's descent angle, rather than rounding up the whole individuals.

L319: I understood that if the ascent angle is too shallow, the time rate of ascent, which is costly, increases, which leads to an increase in COTHD. The ODBA is twice as large at an ascent angle of 0° by the regression line than at any descent angle. The increase of COTHD at the ascent angle of 0° seems to be caused by this assumption. This problem arises from the regression on the quadratic curve. The force due to gravity increases with $\sin \phi$ during ascent. In the range of 0° to 40° , the sine curve can be approximated by linear regression. This is probably because COTHD does not have a convex downward curve in the case of linear regression, but the reasoning for the regression on the quadratic curve needs to be explained. According to the author's explanation, when the ascent angle is shallow, the ascent is caused by the lift of the pectoral fin, which means that the ODBA does not increase?

Review form: Reviewer 2

Is the manuscript scientifically sound in its present form?

Yes

Are the interpretations and conclusions justified by the results?

Yes

Is the language acceptable?

Yes

Do you have any ethical concerns with this paper?

No

Have you any concerns about statistical analyses in this paper?

No

Recommendation?

Accept with minor revision (please list in comments)

Comments to the Author(s)

Dear Dr Andrzejczek,

Thanks for addressing my previous comments on your manuscript. I can see that you have done a lot of work and I think it has noticeably improved the manuscript.

I am happy with all the changes you have made. My only remaining concern with this paper is that you have not made an attempt to make any part of your analysis reproducible. I can see that you use some non-open access software like Igor Pro and perhaps it's not easy to show the series of "clicks" involved in the data processing or analysis you carried out in that environment. However, you also use R and I feel strongly that you need to provide the code you used to fit the statistical models, since R is an open access programming language and environment. It shouldn't be difficult to do that.

You argue in your response that you don't feel it is necessary to provide code for the statistical models fitted in R because GLMMs are part of a standard framework. I would disagree, these are complex models and all of the decisions involved in fitting them are important. At the very least, I think it is important to promote good practice among readers by providing code that supports reproducibility where at all possible.

Given it is a condition of publication in RSOS "that authors make their supporting data, code and materials available", I would argue that this is a critical bit of presenting your work.

I have some very minor editorial suggestions. Well done for this nice piece of work.

Line 17: Double use of "patterns" is a bit repetitive, you could perhaps remove the first one.

Line 22: By low angle do you mean a small angle? If so, I would consider calling it a small angle instead.

Line 163: Do you mean models were fitted? I would use fitted when using existing software.

Line 171: Time series isn't hyphenated

Decision letter (RSOS-200789.R0)

Dear Dr Andrzejczek

On behalf of the Editor, I am pleased to inform you that your Manuscript RSOS-200789 entitled "Depth dependent dive kinematics suggest cost-efficient foraging strategies by tiger sharks" has been accepted for publication in Royal Society Open Science subject to minor revision in accordance with the referee suggestions. Please find the referees' comments at the end of this email.

The reviewers and Subject Editor have recommended publication, but also suggest some minor revisions to your manuscript. Therefore, I invite you to respond to the comments and revise your manuscript.

- Ethics statement

- Data accessibility

If you wish to submit your supporting data or code to Dryad (<http://datadryad.org/>), or modify your current submission to dryad, please use the following link:
<http://datadryad.org/submit?journalID=RSOS&manu=RSOS-200789>

- **Competing interests**

- **Authors' contributions**

- **Acknowledgements**

- **Funding statement**

Because the schedule for publication is very tight, it is a condition of publication that you submit the revised version of your manuscript before 24-Jul-2020. Please note that the revision deadline will expire at 00.00am on this date. If you do not think you will be able to meet this date please let me know immediately.

on behalf of Professor Len Thomas (Associate Editor) and Kevin Padian (Subject Editor)
openscience@royalsociety.org

Associate Editor Comments to Author (Professor Len Thomas):

Associate Editor

Comments to the Author:

Dear Dr Andrzejczek,

Thank-you for re-submitting your manuscript. Both reviewers and I agree that you have addressed their previous comments; the manuscript is now nearly ready for publication.

However, I am returning it one more time to allow you to address further very minor editorial suggestions made by both reviewers and to provide code and data associated with the GLMM analysis, as suggested by Reviewer 2. Please note that it is a condition of publication in RSOS "that authors make their supporting data, code and materials available". You are welcome either to supply it as supplemental materials or to link through to some other archiving site such as Data Dryad.

I look forward to seeing your final submission; I do not anticipate it needing to go out for review again and so expect to be able to accept it quickly upon receipt.

With best wishes, Len Thomas

Reviewer comments to Author:

Reviewer: 1

Comments to the Author(s)

All peer review comments appeared to have been adequately addressed.
I have only a few points to be made.

L309: I understand that there is a lowest angle for gliding, but I don't understand what the lowest possible angle is for tailbeating even during descent. The lowest descent angle where the ODBA is significantly smaller than the level swimming? There seems a unique descent angle for each shark in Fig.S4 and this represents to be the lowest possible angle? If there is a unique descent angle for each shark, the comparison to the model should be made to see if the ascent angle of each shark is consistent with the model that minimizes the COTHD for each shark's descent angle, rather than rounding up the whole individuals.

L319: I understood that if the ascent angle is too shallow, the time rate of ascent, which is costly, increases, which leads to an increase in COTHD. The ODBA is twice as large at an ascent angle of 0° by the regression line than at any descent angle. The increase of COTHD at the ascent angle of 0° seems to be caused by this assumption. This problem arises from the regression on the quadratic curve. The force due to gravity increases with $\sin \phi$ during ascent. In the range of 0° to 40° , the sine curve can be approximated by linear regression. This is probably because COTHD does not have a convex downward curve in the case of linear regression, but the reasoning for the regression on the quadratic curve needs to be explained. According to the author's explanation, when the ascent angle is shallow, the ascent is caused by the lift of the pectoral fin, which means that the ODBA does not increase?

Reviewer: 2

Comments to the Author(s)

Dear Dr Andrzejczek,

Thanks for addressing my previous comments on your manuscript. I can see that you have done a lot of work and I think it has noticeably improved the manuscript.

I am happy with all the changes you have made. My only remaining concern with this paper is that you have not made an attempt to make any part of your analysis reproducible. I can see that you use some non-open access software like Igor Pro and perhaps it's not easy to show the series of "clicks" involved in the data processing or analysis you carried out in that environment. However, you also use R and I feel strongly that you need to provide the code you used to fit the statistical models, since R is an open access programming language and environment. It shouldn't be difficult to do that.

You argue in your response that you don't feel it is necessary to provide code for the statistical models fitted in R because GLMMs are part of a standard framework. I would disagree, these are complex models and all of the decisions involved in fitting them are important. At the very least, I think it is important to promote good practice among readers by providing code that supports reproducibility where at all possible.

Given it is a condition of publication in RSOS "that authors make their supporting data, code and materials available", I would argue that this is a critical bit of presenting your work.

I have some very minor editorial suggestions. Well done for this nice piece of work.

Line 17: Double use of "patterns" is a bit repetitive, you could perhaps remove the first one.

Line 22: By low angle do you mean a small angle? If so, I would consider calling it a small angle instead.

Line 163: Do you mean models were fitted? I would use fitted when using existing software.

Line 171: Time series isn't hyphenated

Author's Response to Decision Letter for (RSOS-200789.R0)

See Appendix C.

Decision letter (RSOS-200789.R1)

Dear Dr Andrzejczek,

It is a pleasure to accept your manuscript entitled "Depth dependent dive kinematics suggest cost-efficient foraging strategies by tiger sharks" in its current form for publication in Royal Society Open Science.

on behalf of Professor Len Thomas (Associate Editor) and Kevin Padian (Subject Editor)
openscience@royalsociety.org

Appendix A**ROYAL SOCIETY
OPEN SCIENCE****Depth dependent dive kinematics suggest cost-efficient
foraging strategies by tiger sharks**

Journal:	Royal Society Open Science
Manuscript ID	RSOS-191258
Article Type:	Research
Date Submitted by the Author:	22-Jul-2019
Complete List of Authors:	Andrzejaczek, Samantha; University of Western Australia, Oceans Graduate School & The UWA Oceans Institute; Australian Institute of Marine Science, ; Stanford University, Hopkins Marine Station Gleiss, Adrian; Murdoch University, Centre for Sustainable Aquatic Ecosystems, Harry Butler Institute Lear, Karissa; Murdoch University, Centre for Sustainable Aquatic Ecosystems, Harry Butler Institute Pattiaratchi, Charitha; University of Western Australia, Oceans Graduate School & The UWA Oceans Institute Chapple, Taylor; Stanford University, Hopkins Marine Station Meekan, Mark; Australian Institute of Marine Science,
Subject:	behaviour < BIOLOGY, ecology < BIOLOGY, biomechanics < BIOLOGY
Keywords:	biologging, vertical movement, cost of transport, top predator, ODBA, movement ecology
Subject Category:	Biology (whole organism)

Author-supplied statements

Relevant information will appear here if provided.

Ethics

Does your article include research that required ethical approval or permits?:

Yes

Statement (if applicable):

All methods used were in accordance with approved guidelines by the University of Western Australia Animal Ethics Committee (RA/3/100/1437), and under permit numbers 2881 (WA Department of Primary Industries and Regional Development) and 08-000322-3 (WA Department of Biodiversity, Conservation and Attractions).

Data

It is a condition of publication that data, code and materials supporting your paper are made publicly available. Does your paper present new data?:

Yes

Statement (if applicable):

Data can be accessed at <https://datadryad.org/review?doi=doi:10.5061/dryad.n3r1680>

Well done for providing the data but without the code used for data preparation and statistical analyses it's not possible to reproduce your approach.

Conflict of interest

I/We declare we have no competing interests

Statement (if applicable):

CUST_STATE_CONFLICT :No data available.

Authors' contributions

This paper has multiple authors and our individual contributions were as below

Statement (if applicable):

SA, MM, and AG conceived the study. SA, AG, TC, and KL performed fieldwork/data collection. SA, AG, CP, and MM analyzed and interpreted the data, and SA led the writing of the manuscript. All authors contributed to drafting the article, and reading and approving the final manuscript.

1 Depth dependent dive kinematics suggest cost-efficient 2 foraging strategies by tiger sharks

**Samantha Andrzejczek^{1,2*}†, Adrian C. Gleiss³, Karissa Lear³, Charitha Pattiaratchi¹, Taylor
Chapple⁴ and Mark Meekan²**

¹Oceans Graduate School & The UWA Oceans Institute, The University of Western Australia, Crawley, WA, 6009, Australia

²The Australian Institute of Marine Science, Crawley, WA, 6009, Australia

³Centre for Sustainable Aquatic Ecosystems, Harry Butler Institute, Murdoch University, Murdoch, WA, 6150, Australia

⁴Hopkins Marine Station, Stanford University, Pacific Grove, CA, 93950, USA

*Author for correspondence: sandrzejczek@gmail.com

†Present address: Hopkins Marine Station, Stanford University, Pacific Grove, CA, 93950, USA

Abstract (<200 words)

Tiger sharks *Galeocerdo cuvier* are a keystone, top-order predator that are assumed to engage in cost-
efficient movement and foraging patterns. To investigate the extent to which patterns of oscillatory
diving by these animals conform to a strategy of cost-efficient foraging, we used a biologging approach
to model their cost of transport. High-resolution biologging tags with tri-axial sensors were deployed
on 21 tiger sharks at Ningaloo Reef for durations of 5-48 hours. Using overall dynamic body
acceleration (ODBA) as a proxy for energy expenditure, we modelled the cost of transport of
oscillatory movements of varying geometries in both horizontal and vertical planes for tiger sharks.
The cost of horizontal transport was minimized at absolute dive angles greater than 0°, so that
oscillations conserved energy relative to swimming at a level depth. Optimization of vertical travel
occurred at steeper angles. The absolute dive angles of tiger sharks increased between inshore and
offshore zones, presumably to reduce energy costs while continuously hunting for prey in both benthic
and surface habitats. Oscillatory movements of tiger sharks conform to strategies of cost-efficient
foraging, and shallow sandflats appear to be an important habitat for both hunting prey and saving
energy.

**Keywords:** biologging; vertical movement; cost of transport; top predator; ODBA

This is confusing, I have made more comments below. Where they optimal even at 1 degree? What does this mean biologically? Is there any uncertainty associated with this conclusion?

Introduction

The tiger shark is a top-order predator that has wide-ranging impacts on the ecology of the tropical
 and warm-temperate marine ecosystems it inhabits [1, 2]. These sharks are generalist feeders with
 anatomical specializations for feeding on large prey [3-7]. Although they actively hunt prey, they are
 also facultative scavengers [7], which avoids or reduces many of the energetic costs involved in the
 process of active predation [7, 8]. However, injured or dead prey are windfalls that only occur
 sporadically and for the most part, tiger sharks must actively locate, hunt and capture their food. Like
 all top-order predators, they are assumed to do so in a cost-efficient manner [9].

Tagging studies have revealed that the fine-scale movements of tiger sharks are characterised by
 patterns of oscillatory ascents and descents (“dives”) throughout the water column [2, 10], a behavior
 that is shared by many species of large epipelagic fishes [11]. For tiger sharks, these dives have been
 recorded in a number of habitats, ranging from very shallow (<4 m) seagrass beds in Shark Bay [2], to
 deeper (>100 m), offshore waters in Hawaii and Brazil [10, 12, 13]. It has been argued that these
 oscillatory movements are not likely to be primarily driven by behavioural thermoregulation, because
 they often occur in well-mixed and/or shallow environments [2, 10] where there is little gradient in
 temperature between the surface and the seafloor. Instead, previous studies have suggested that
 oscillatory diving provides an effective search strategy for an animal that feeds on prey both at the
 surface and near the seabed [2, 10]. More recently, tiger sharks tagged at Ningaloo Reef have been
 found to make oscillatory dives during the tortuous movements associated with area-restricted
 search, though these oscillatory dives also occurred throughout directional swimming, suggesting that
 factors other than foraging also contribute to these patterns [14].

In addition to allowing sharks to search for prey, oscillatory dives may represent a strategy to reduce
 the cost of transport. Weihs [15] proposed that a two-stage mode of swimming may allow negatively-
 buoyant animals to reduce energy expenditure relative to swimming at a level depth between two
 horizontal points, whereby individuals oscillate throughout the water column, gliding on descent and
 actively swimming on ascent. Recent advances in the use of tri-axial sensors in biologging tags provide
 a means to test this hypothesis in a natural setting. Tri-axial accelerometers can be used to calculate
 dive angles in addition to overall dynamic body acceleration (ODBA), a parameter that can act as a
 proxy for energy consumption [16], and can therefore be used to quantify and compare the energy
 costs of different movement behaviours. Gleiss *et al.* [17] used a biologging approach to explore how
 differences in the dive geometry in whale sharks, *Rhincodon typus*, influenced the cost of transport
 with respect to both horizontal and vertical distance, using dynamic body acceleration as a proxy for
 power. Empirical optimality models suggested that some dive profiles reduced the cost of horizontal

This isn't a good reference for general cost-efficiency of movement in top predators. The paper is about obligate scavengers.

Does this still count as a “dive”? How do you scale the definition with available depth?

Has area-restricted search behaviour been shown to be more strongly linked with foraging than directed movements? If not, there is no way of knowing whether hunting behaviour is linked to movement patterns.

transport, while other steeper-angled dives minimized the cost of vertical transport; indicating that
 the choice of dive undertaken by an individual may be dependent on the ecological context i.e. travel,
 search or foraging [18].

As tiger sharks are negatively buoyant [10], we hypothesise that they may also exploit their weight in
 water and use oscillatory movements to reduce the cost of transport while travelling or searching for
 prey. However, given that **dead prey will sink to the bottom** whereas active prey may be most easily
 ambushed at the surface where it can be approached from below [19], movement strategies may be
 habitat dependent. Tiger sharks tagged at Ningaloo Reef were found to undertake oscillatory
 movements in both inshore and offshore habitats [14]. In very shallow habitats (<3 m), tiger sharks
 would have no need to undergo vertical search patterns given the short distance between the surface
 of the water and the seabed. In deeper habitats, however, individuals would need to actively transit
 through a larger water column between the surface and seabed in their search for prey. We therefore
 predict that the diving energetics and kinematics of these animals will be dependent on the depth of
 the habitat being occupied, with deeper waters featuring steeper dive angles when searching for prey
 to allow for **faster transit times**.

This depends on the type of prey. Dead whales will float, for example.

I'm a bit confused here, do they search while changing depth? Or do they also hunt while changing depth? It's not clear that it would always be better to do this as fast as possible

Here, we use a biologging approach to examine the energetics of oscillatory diving behavior in tiger
 sharks at Ningaloo Reef, Western Australia. We examine the incidence of gliding behaviour in tiger
 sharks and model the cost of transport of oscillatory movements of varying movement geometries
 using ODBA as a proxy for energy expenditure. We document the range of dive angles utilized by
 tagged sharks, particularly in relation to habitat depth, discuss the role of oscillatory movements in
 both efficient foraging and energy conservation, and re-evaluate the drivers of oscillatory diving in this
 species.

Materials and methods

**Data collection:**

How many sharks were tagged? I know you have table in S1 but it would be good to state it here too

**Tiger sharks** were captured and tagged at Ningaloo Reef, Western Australia in April and May 2017
 following the methods described in Andrzejczek *et al.* [14]. In brief, tiger sharks were captured using
 baited drumlines and secured alongside a 5.8 m vessel with the leader and tailrope. Biologging tags
 were then clamped to the dorsal fins for periods of 7-48 hours (see table S1). Tags were equipped with
 tri-axial accelerometers, magnetometers and gyroscopes, and sensors for depth, temperature and
 light. All sensors recorded continuously at 20 Hz. In addition, 14 of the 22 deployments recorded video
 at pre-programmed hours of the day for a maximum of six hours per deployment. The tags detached

from the clamp in the days following tagging, and were recovered using a handheld VHF receiver
 operated from a vessel.

**Data processing:**

Data recorded by the tags were processed to obtain a number of parameters classifying shark
 movements, behaviour and the external environment. Vertical phases of movement, dive angles,
 overall dynamic body acceleration (ODBA), and tailbeat kinematics were calculated as described in
 Andrzejaczek *et al.* [14]. In brief, the depth record was split into vertical phases of ascent, descent and
 constant depth. Tri-axial sensor data were then used to calculate the pitch angles of vertical
 movements, ODBA, and the signal amplitude and frequency of tailbeats in Igor Pro ver. 7.0.4.1
 (Wavemetrics, Inc. Lake Oswego, USA). Tailbeat data were used to calculate the recovery period from
 capture following methods outlined by Whitney *et al.* [20]. Video data were also processed as per
 Andrzejaczek *et al.* [14], and were used to record interactions with prey, determine habitat types, and
 validate behaviours recorded by the sensors (e.g. tailbeats and gliding behaviour). In addition, sensor
 data were used to compute gliding behaviour, and ascent and descent speed, as described below.

*Gliding behaviour*

This is well described - more of this!

We used a continuous wavelet transformation on the dynamic component of the sway (i.e. lateral)
 axis to calculate the signal amplitude and frequency of shark tailbeats using the angular velocity data
 [14, 21]. These data were used to quantify the incidence of gliding behaviour – defined here as a
 cessation of tailbeats for more than one second – through a two-step process as per Andrzejaczek *et*
 *al.* [22]. Briefly, (1) gliding behaviour was isolated using the ‘k-means cluster’ function in the
 Ethographer for Igor Pro [21]. This function clustered the spectra computed by the wavelet
 transformation based on similarity of shape. The behavioural spectrum with the lowest peaks in
 angular velocity signal amplitude was assumed to represent gliding behaviour [10], and the incidence
 of the resulting cluster was then inspected against the dynamic sway data. As this cluster did not
 match with gliding behaviour in some individuals (i.e. tailbeats evident in sway data were classified to
 be gliding, and vice versa), (2) a mask was created using the characteristics of the angular velocity
 signal amplitude and tailbeat frequency where the cluster was judged to correctly classify gliding
 behaviour (from visual inspection of the dynamic sway data and concurrent videos). This mask was
 then used to extract glides from all sharks.

It's not clear what this is describing, what is a mask? Many of your readers might not know. I know you refer to an already published paper [22] but if you are going to describe the behavioural classification routine, even briefly, it needs to be clear.

*Ascent and descent speeds*

Vertical velocity (VV), defined as the rate of change in depth over a one second period, and pitch (φ)
 were used to estimate the mean speed of ascents and descents through trigonometry as per:

$$\text{Speed (ms}^{-1}\text{)} = \frac{\text{Vertical velocity (ms}^{-1}\text{)}}{\sin(\varphi)}$$

This, however, could only be calculated when pitch exceeded 20° due to the large errors associated
 with estimating speed at low pitch angles [17].

What is the uncertainty associated with depth?
 This will directly affect pitch angle.

*Window size and statistics*

The sampling window used for analysis was determined by calculating the time period where the
 highest variance in turning angles was observed, while being of sufficient size to capture the longest
 recorded dives in their entirety as per Andrzejczek *et al.* [14]. This time window was estimated to be
 15 minutes (900 seconds). Therefore, a number of vertical movement parameters were summarized

Does this make
 it highly
 dependent on
 this particular
 dataset?

for each 15 minute window of each deployment including mean (\pm standard deviation) and maximum
 depth, ascent pitch, descent pitch, ascent VV and descent VV. The percent of time spent moving
 vertically (ascending and descending), termed the 'diving ratio', was also calculated within each
 window for each individual as per:

Did you allow for any kind of random variability or error around a depth measurement or was
 any change in depth considered as "moving vertically"? Does a tiger shark ever maintain
 depth without any variability in depth? What was the accuracy of the depth sensor?

$$\text{Diving ratio} = \frac{\text{Time vertically moving in window (seconds)}}{\text{Total time in sampling window (900 seconds)}}$$

Data analysis:

*Generalised linear mixed models (GLMMs)*

Why do you need a mixed model for this? Did you just have
 a random intercept or also random slopes?
 It would be good to add a sentence explaining the rationale.

Generalised linear mixed models (GLMMs) were built using the nlme package in R 3.4.0 [23, 24] to
 investigate possible relationships between seabed depth and vertical movement behaviours in tiger
 sharks. The maximum depth (m) recorded within each time window was used as a proxy for seabed
 depth (based on video analysis; see Andrzejczek *et al.* [14]) and was set as the explanatory variable,

was this the only explanatory variable?

and tiger shark identity set as a random variable for all models. Ascent pitch, descent pitch, ascent VV,
 descent VV and diving ratio were all set sequentially as response variables. We used the corAR1

function to account for temporal auto-correlation in our datasets [25]. Together with nautical charts
 from Ningaloo Reef, maximum depth was used to classify windows as either 'inshore' (<25 m, inside
 the reef) or 'offshore' (>25 m, outside the reef). GLMMs were analysed separately for inshore and
 offshore periods due to heterogeneity in residuals and an unbalanced design. The resulting models

Windows in what? This comes across as vague here

were compared against the null models and ranked using Akaike's information criterion (AIC).

What kind of
 temporal
 autocorrelation
 does this
 function
 assume? Did
 you look for
 what kind of
 autocorrelation
 was present in
 the data? What
 are the
 consequences
 of mis-
 specifying the
 shape of the
 autocorrelation
 ?

To investigate if observed changes in diving ratio with depth were an artifact of our selected sampling
 window, we calculated diving ratio for oscillations occurring in increasingly deeper water given a fixed

What were your
 candidate models? How
 did you chose them?

Do you mean unequal variance in the residuals with depth? Heterogeneity is perhaps harder to understand, you
 could add it in brackets.

Why is it bad to have an unbalanced design? I think it's worth mentioning bias here.
 Both of these statements could use a little bit of context. One or two sentences would really help a lot.

This section is confusing, again, I think some context would help.

interval of level swimming at the surface and seabed. One-hour long depth traces were simulated for
 a hypothetical shark oscillating in depths of 5, 10, 20, 30, 40, 50 and 60 m. The ascent VV and descent
 VV for each depth zone were determined following relationships calculated between VV and seabed
 depth (see above). The fixed interval spent at the surface and on the seabed between vertical
 movements was set at two minutes following exploration of the depth traces. Diving ratio was
 calculated for each of the four 15 minute windows, and averaged for the hour so that one value of
 diving ratio was calculated for each depth. Where do the four 15min windows come from?

*Cost of transport models*

We modelled the cost of transport of oscillatory movements of varying geometries in relation to
 optimization of either horizontal or vertical distance travelled following methods similar to those
 described by Gleiss *et al.* [17] (figure 1). First, we calculated the total mechanical cost (TC) of an
 oscillation (an ascent (a) and descent (d) combined) in units of ODBA (g) using the equation:

$$166 \quad TC = T_a \times ODBA_a + T_d \times ODBA_d + k \times (T_d + T_a)$$

The average ODBA for descent and ascent? What qualified as a descent and ascent? You define these below but it would be good to bring them up.

Where T_a and T_d are the time spent ascending and descending, respectively, $ODBA_a$ and $ODBA_d$ are the
 ODBA of ascents and descents, respectively, and k is a proxy for basal metabolic cost. Previous studies
 have shown that basal metabolic costs are approximately 60% of routine metabolic rate in sharks [see
 26 and references therein], and therefore we estimated k at 60% of the mean ODBA recorded for all
 sharks (0.026 g) or $k = 0.0156$ g. T_a , T_d , $ODBA_a$, and $ODBA_d$, were all calculated depending on pitch
 angle (φ). $ODBA_a$ was estimated from the relationship between ODBA and φ_a (figure 2A), and for
 $ODBA_d$, a single mean value of ODBA during descents (0.012 g) was used, as no relationship was found
 between φ_d and ODBA (figure 2B). T_a and T_d are a function of φ_a and φ_d , respectively, depth, and mean
 speed, and were calculated using the following equations:

How did this compare to values from the literature for tiger sharks or other species of shark?

$$176 \quad T_a = \frac{\frac{Depth}{\sin(\varphi_a)}}{\text{mean ascent speed}}$$

$$177 \quad T_d = \frac{\frac{Depth}{\sin(\varphi_d)}}{\text{mean descent speed}}$$

We used a fixed mean estimate of speed for both ascents (0.87 m s^{-1}) and descents (0.85 m s^{-1}) as no
 relationship was found between speed and pitch angle. TC was calculated for fixed ascent angles from
 5° to 45° , binned in 5° increments. Why did you bin the pitch angle? For each bin of ascent angles, TC was calculated sequentially for
 decent angles of 5° to 20° , binned in 5° increments.

We constructed two different models describing the cost of horizontal transport (COT_{HD}) and cost of
 vertical transport (COT_{VD}) for tiger sharks. These models calculated the cost of moving a unit of
 horizontal (HD) and vertical distance (VD) respectively, and were used to determine the angles that
 optimized the efficiency of transport on each of these scales. The COT_{HD} was modelled by:

$$COT_{HD} = \frac{TC}{HD}$$

Where horizontal distance was calculated from ascent and descent pitch using the equation:

$$HD = \frac{Depth}{\tan(\phi_d)} + \frac{Depth}{\tan(\phi_a)}$$

The COT_{VD} was modelled by:

$$COT_{VD} = \frac{TC}{2 \times Depth}$$

All model calculations used oscillations of 10 m depth, however, the resulting COT for horizontal and
 vertical distance was the same regardless of depth. Did you have any sharks moving at or very close to the surface?
 Is there higher turbulence at the surface which might increase cost of transport?

**Results** You talk about this below, is it relevant to mention it here too?

**Track summary**

It would still be useful to know how many tags were deployed to begin with

196 Tag data was recovered from a total of 21 tiger sharks (ranging 2.65 – 3.8 m total length) tagged at
 197 Ningaloo Reef during this study (see table S1 for full summary of tag deployments). We assumed a
 198 recovery period from capture and tagging by the sharks of four hours based on an analysis of tailbeats,
 and for this reason the first four hours of each dataset were excluded from further analysis [14].
 Evidence for recovery after four hours was also provided by the video records, which showed
 investigation of prey and consumption of a discarded fish head by sharks within two hours of release
 after tagging.

What is the measure of error you give here? $11.6 - 17.5 = -5.9$ which would be above the surface

Tagged tiger sharks swam at a mean depth of 11.6 ± 17.5 m, predominately residing in inshore
 habitats, with four tiger sharks moving into offshore habitats and one individual diving to a maximum
 depth of 94 m. Sharks moved vertically for a mean of $38.4 \pm 26\%$ of their track and utilised significantly
 steeper descent ($-11.11 \pm 3.6^\circ$) than ascent ($9.38 \pm 2.6^\circ$) angles (Wilcoxon signed-ranks test, $P < 0.001$).

**Gliding behaviour** I would argue that this is the main result of your study. There is a lot of work that has gone into dive angles
 and body density in air-breathing divers, e.g. Aoki et al. 2011 Journal of Experimental Biology. This aspect
 of your work is a generally interesting result, beyond tiger sharks in your study area.
 Sorry to sound negative but the claims about energy expenditure inshore and offshore as a function of
 dive angle are less convincing, the way they are currently presented.

There was a high level of individual variation in the proportion of gliding descents by tiger sharks
 (range 0-35%, figure 3). Six sharks glided for more than 19% of their total descent time, of which three
 glided for more than 30% of their descents. A maximum uninterrupted glide time of two minutes was
 recorded for one individual. Gliding was rarely recorded in the first four hours of each tag deployment,
 (subsequently designated as the recovery period), and no gliding behaviour was recorded for two
 sharks (figure 3). **Gliding was exhibited by tiger sharks in both inshore and offshore zones and occurred**
 **at a minimum descent angle of -8° and at a mean angle of $-12.2 \pm 2.7^\circ$.**

Can you say something about body density from these glides?

GLMMs: Is there more than one inshore and one offshore model? How come? It would be good to say how many models you fit and how they were structured

**Relationship between seabed depth and vertical movement behaviours**

Do you mean biologically significant, or statistically significant? If the latter, at what level?

GLMMs revealed **significant** relationships between seabed depth and vertical movements of tiger
 sharks (table 1). For inshore **GLMMs**, all models retained their predictor variables of vertical
 movement, whereas for offshore models, only the random effect of tiger shark identity was retained.
 Up to depths of 25 m, diving ratio, pitch and VV all increased with seabed depth, after which point
 relationships plateaued and displayed high levels of variation (figure S1). Inshore, maximum/seabed
 depth explained 35% (marginal R^2) of the variation in both diving ratio and descent pitch. For all
 models, high conditional R^2 values (up to 55%) suggested high inter-individual variability in vertical
 movements (table 1). Mean absolute pitch angles increased between inshore and offshore habitats
 from $10.3 \pm 2.8^\circ$ to $14.1 \pm 4.7^\circ$ for descent, and 9 ± 2.3 to $10.7 \pm 3.2^\circ$ for ascent (figure 4).

The mean diving ratio increased from $31 \pm 23\%$ to $69 \pm 17\%$ between inshore and offshore habitats
 (figure 5). The simulated depth data generated using VV from GLMM relationships and fixed intervals
 of level swimming between vertical movements resulted in diving ratios that were similar to those
 observed in the raw data (figure 5). Despite VV also increasing with depth, diving ratio increased
 quickly between 5 and 20 m depths, after which the slope rapidly decreased with increasing depth.

**Cost of transport models**

What made you chose a one-way ANOVA? Was it not possible for the cost of ascent to be smaller than that of descent? In general, it's always better to run a two-way ANOVA to allow for a fair comparison, unless it's really impossible for an effect to go in both directions. That doesn't seem to be the case here.

The mechanical cost of ascending was greater than that for descending (**one-way ANOVA, $F_{1,240002} =$**
 **$1635, p < 0.0001$**). The cost of level swimming by tiger sharks could not be calculated as the effects of
 **See comment above**
 **wave action at the surface frequently created superficially high ODBA** levels that could not be removed
 reliably from the data. These effects of surface waves were confirmed by the video data.

For all sharks, a positive quadratic relationship was found between mean binned ascent angle and
 ODBA (figure 2A, $r^2 = 0.96$). This relationship had positive slopes for all individual sharks with a mean
 r^2 of 0.8 ± 0.23 (range 0.12-0.99). No consistent relationship was found between mean binned descent
 angle and ODBA (figure 2B). We regressed swimming speed and body pitch where pitch was greater
 than 20° for each individual shark, and did not detect quantifiable changes in speed with pitch angle

(table S2). Where the relationship was found to be significant, slope did not exceed 0.02, and we
 therefore concluded that increased locomotory activity at increasing pitch was due to counteracting
 gravity, rather than an inherent increase in speed with pitch angle [17].

We used the relationship between ascent pitch angle and ODBA to calculate ODBA as a proxy for
 mechanical cost. This was done at 5° intervals for COT models, and a single mean value of ODBA was
 used for descents. As gliding behaviour was highly variable and we found no evidence of accelerating
 VV towards the maximum depth of dives, we did not calculate the terminal velocity as per Gleiss et al.
 (2011). The total cost of an oscillation of given ascent and descent angles was then standardized by
 the corresponding vertical and horizontal distances travelled in order to calculate the cost of transport
 on each respective plane. We found that the cost of travel was optimized on both horizontal and
 vertical planes at absolute ascent and descent angles greater than 0° (figure 6A). The cost of transport
 for tiger sharks in the horizontal plane was minimized by descending at the lowest possible angle, and
 ascending at an angle of 5-14°; decreasing with a shallower descent angle. The cost of transport in the
 vertical plane was optimized at steeper ascent angles of 35°, and decreased with steeper descent
 angles, however, the potential energy savings at absolute ascent and descent angles steeper than 20°
 and 15° respectively were minimal (figure 6B).

I don't think you are able to make this claim. You haven't measured the profitability of different environments so you can only say there was a higher energy expenditure in certain habitats, but the net gain could have been much higher, if the profitability was much higher too.

I also find it quite a confusing statement. Maybe I've missed the point, but it comes across like you are saying that they optimise diving behaviour when they are diving, since they can only have an absolute swim angle greater than zero when they are diving. Can you make it clearer?

Discussion

Our study provides evidence that oscillatory movements through the water column allowed tiger
 sharks to conserve energy while travelling and searching for prey and result in a strategy of cost-
 efficient foraging. Our empirical models of the cost of transport showed that, while in transit, tiger
 sharks required less energy to move in oscillatory dives at shallow angles through the water column
 than they did while swimming at a constant depth. Shallow angles of ascent and descent allowed tiger
 sharks to reduce their cost of transport in the horizontal plane, whereas steeper angles reduced their
 energetic costs in the vertical plane. The absolute dive angles of tiger sharks increased between
 inshore and offshore zones, presumably to reduce energy costs while continuously hunting for prey
 in both benthic and surface habitats.

This is a much better, less provocative way of framing your results. Optimality is a thorny issue that will always raise eyebrows.

This is confusing, I can't tell what you mean here

Cost of vertical movements:

Patterns of gliding varied among tagged sharks. Gliding occurred through 0-35% of the overall descent
 phase of oscillations for individuals at Ningaloo, a finding comparable to that of Nakamura *et al.* [10].
 Although the majority of sharks displayed increasing levels of gliding behaviour post-release, as
 previously observed in blacktip sharks [20], some sharks – including an individual tagged for >15 hours
 - recorded no gliding behavior throughout tag deployment. This variability may be driven by minor

272 individual differences in body composition [27, 28], and/or behavioural differences in their response
273 to catch and release. At all spatial scales movements of tiger sharks have been characterized by high
274 levels of individual variation [29, 30].

Interesting

[revised manuscript text omitted]

**Seascape effects on energy expenditure and gain**

Our results suggest that very shallow waters (<3 m) are the most cost-efficient place for tiger sharks
to forage, and may therefore represent an important energy landscape for these animals. Variation in

an energy landscape occurs where the cost of transport is influenced by physical characteristics of the
environment [32]. This results in animals modifying movements among habitats to either exploit
opportunities for energy gain, and/or reduce costs [33]. Establishing the variability in movement costs
across landscapes can help us to understand how and why animals distribute themselves in space [32].
For tiger sharks, alternating between the seabed and surface in the search for prey means that the
energetic costs of foraging increase with depth. Very shallow environments (<3 m), such as the
sandflats of Ningaloo Reef, represent a landscape where tiger sharks can move and forage at the most
cost-efficient dive angles. In addition, pseudo-tracks, video data, and observations from the
ecotourism industry have highlighted the importance of the shallow sandflat environments for tiger
sharks at Ningaloo Reef [14]. In these habitats, tiger sharks often displayed the tortuous movement
patterns thought to denote foraging behavior [34], and predator-prey interactions have frequently
been observed here. Tiger sharks tagged in other regions have also been found to display a preference
for shallow habitats, including seagrass habitats less than four meters in depth in Shark Bay, and
lagoon habitats in Hawaii [35, 36].

This a tricky point because the habitat essentially enforces shallow dive angles, so to what extent can you say this is a strategy? In addition, how long is an average tiger shark? If they are in the range of 2-3 metres in length then there is a hard limit to the angle they can form

The preferential selection of these shallow water habitats has implications for the distribution of
predation pressure, and therefore the distribution of prey species. A study conducted over 15 years
in the largely seagrass habitat of Shark Bay, Western Australia, found that the presence of tiger sharks
influenced the movements and behaviour of a number of prey species, including dolphins, turtles and
dugongs [37]. Given the prolonged residency of tiger sharks at Ningaloo Reef (several months; [30]) it
seems likely that they will have similar structuring roles in these coral reef environments.

What do you mean here?
Can you justify and explain this claim?

**Conclusion**

Collectively, our data showed that the oscillatory movements of coastal tiger sharks are the product
of a combined foraging and energy conservation strategy. Tiger sharks modified their vertical
movement behaviours between inshore and offshore zones, presumably to optimize the use of energy
while continuously hunting for benthic and air-breathing prey. The shallow sandflats of Ningaloo Reef
appear to be an important habitat for hunting prey and saving energy.

I feel like this is an overly bold claim. The balance of energy is made up of both inputs and outputs. You have only measured outputs. What if deeper areas are much more profitable than sandflats? You don't mention this at all.

Acknowledgements:

SA was supported by an Australian Government Research Training Program (RTP) Scholarship and an
UWA top-up scholarship. We thank Murdoch University and Coral Bay Research Station for
accommodation and vessel use while conducting fieldwork. Additional logistical support was provided
by Olwyn Hunt and Louise Scott from the Australian Institute of Marine Science. This work would not
have been possible without the generous support of numerous volunteers, particularly: Frazer

McGregor, Abraham Sianipar, Adam Jolly, Blair Bentley, Evan Byrnes, Garry Teesdale and Michael
Tropiano providing valuable field support; Olivia Seeger and Abraham Sianipar who helped with the
video analysis. Nikolai Liebsch provided invaluable assistance with tag functioning. Thanks to Lauren
Peel for helpful comments on an earlier version of this manuscript.

11 373 Ethical statement

All methods used were in accordance with approved guidelines by the University of Western Australia
Animal Ethics Committee (RA/3/100/1437), and under permit numbers 2881 (WA Department of
Primary Industries and Regional Development) and 08-000322-3 (WA Department of Biodiversity,
Conservation and Attractions).

21 378 Funding:

Fieldwork and tags were funded by crowdfunding on the Experiment platform (DOI: 10.18258/7190),
a Holsworth Wildlife Research Endowment, a UWA Graduate Research School fieldwork award,
BigWave Productions and the Australian Institute of Marine Science.

29 382 Data accessibility:

On acceptance of manuscript data will be accessible from the Dryad repository. Prior to acceptance,
data can be accessed from <https://datadryad.org/review?doi=doi:10.5061/dryad.n3r1680>

36 37 385 Competing interests:

38
39
40 386 The authors declare no competing or financial interests.

41 42 43 387 Author contributions:

SA, MM, and AG conceived the study. SA, AG, TC, and KL performed fieldwork/data collection. SA, AG,
CP, and MM analyzed and interpreted the data, and SA led the writing of the manuscript. All authors
contributed to drafting the article, and reading and approving the final manuscript.

52 391 Reference List:

1 Heithaus, M. R., Frid, A., Wirsing, A. J., Dill, L. M., Fourqurean, J. W., Burkholder, D., Thomson, J.,
Bejder, L. 2007 State-dependent risk-taking by green sea turtles mediates top-down effects of tiger
shark intimidation in a marine ecosystem. *Journal of Animal Ecology*. **76**, 837-844. (10.1111/j.1365-
2656.2007.01260.x)

2 Heithaus, M., Dill, L., Marshall, G., Buhleier, B. 2002 Habitat use and foraging behavior of tiger
sharks (*Galeocerdo cuvier*) in a seagrass ecosystem. *Marine Biology*. **140**, 237-248. (10.1007/s00227-
001-0711-7)
3 Lowe, C. G., Wetherbee, B. M., Crow, G. L., Tester, A. L. 1996 Ontogenetic dietary shifts and
feeding behavior of the tiger shark, *Galeocerdo cuvier*, in Hawaiian waters. *Environmental Biology of*
*Fishes*. **47**, 203-211. (10.1007/bf00005044)
4 Heithaus, M. R. 2001 The biology of tiger sharks, *Galeocerdo cuvier*, in Shark Bay, Western
Australia: sex ratio, size distribution, diet, and seasonal changes in catch rates. *Environmental*
*Biology of Fishes*. **61**, 25-36. (10.1023/A:1011021210685)
5 Simpfendorfer, C., Goodreid, A., McAuley, R. 2001 Size, sex and geographic variation in the diet of
the tiger shark, *Galeocerdo cuvier*, from Western Australian waters. *Environmental Biology of Fishes*.
**61**, 37-46. (10.1023/a:1011021710183)
6 Ferreira, L. C., Thums, M., Heithaus, M. R., Barnett, A., Abrantes, K. G., Holmes, B. J., Zamora, L. M.,
Frisch, A. J., Pepperell, J. G., Burkholder, D., *et al.* 2017 The trophic role of a large marine predator,
the tiger shark *Galeocerdo cuvier*. *Scientific Reports*. **7**, 7641. (10.1038/s41598-017-07751-2)
7 Hammerschlag, N., Bell, I., Fitzpatrick, R., Gallagher, A. J., Hawkes, L. A., Meekan, M. G., Stevens, J.
D., Thums, M., Witt, M. J., Barnett, A. 2016 Behavioral evidence suggests facultative scavenging by a
marine apex predator during a food pulse. *Behavioral Ecology and Sociobiology*. 1-12.
(10.1007/s00265-016-2183-2)
8 Clua, E., Chauvet, C., Read, T., Werry, J. M., Lee, S. Y. 2013 Behavioural patterns of a tiger shark
(*Galeocerdo cuvier*) feeding aggregation at a blue whale carcass in Prony Bay, New Caledonia.
*Marine and Freshwater Behaviour and Physiology*. **46**, 1-20. (10.1080/10236244.2013.773127)
9 Ruxton, G. D., Houston, D. C. 2004 Energetic feasibility of an obligate marine scavenger. *Marine*
*Ecology Progress Series*. **266**, 59-63.
10 Nakamura, I., Watanabe, Y. Y., Papastamatiou, Y. P., Sato, K., Meyer, C. G. 2011 Yo-yo vertical
movements suggest a foraging strategy for tiger sharks *Galeocerdo cuvier*. *Marine Ecology Progress*
*Series*. **424**, 237-246. (10.3354/meps08980)
11 Andrzejaczek, S., Gleiss, A. C., Pattiaratchi, C. B., Meekan, M. G. 2019 Patterns and drivers of
vertical movements of the large fishes of the epipelagic. *Reviews in Fish Biology and Fisheries*.
(10.1007/s11160-019-09555-1)
12 Afonso, A. S., Hazin, F. H. 2014 Post-release survival and behavior and exposure to fisheries in
juvenile tiger sharks, *Galeocerdo cuvier*, from the South Atlantic. *Journal of Experimental Marine*
*Biology and Ecology*. **454**, 55-62. (10.1016/j.jembe.2014.02.008)
13 Holland, K., Wetherbee, B., Lowe, C., Meyer, C. 1999 Movements of tiger sharks (*Galeocerdo*
*cuvier*) in coastal Hawaiian waters. *Marine Biology*. **134**, 665-673. (10.1007/s002270050582)
14 Andrzejaczek, S., Gleiss, A. C., Lear, K. O., Pattiaratchi, C. B., Chapple, T. K., Meekan, M. G. 2019
Biologging Tags Reveal Links Between Fine-Scale Horizontal and Vertical Movement Behaviors in
Tiger Sharks (*Galeocerdo cuvier*). *Frontiers in Marine Science*. **6**, (10.3389/fmars.2019.00229)
15 Weihs, D. 1973 Mechanically efficient swimming techniques for fish with negative buoyancy.
*Journal of Marine Research*. **31**, 194-209.
16 Gleiss, A. C., Wilson, R. P., Shepard, E. L. 2011 Making overall dynamic body acceleration work: on
the theory of acceleration as a proxy for energy expenditure. *Methods in Ecology and Evolution*. **2**,
23-33. (10.1111/j.2041-210X.2010.00057.x)
17 Gleiss, A. C., Norman, B., Wilson, R. P. 2011 Moved by that sinking feeling: variable diving
geometry underlies movement strategies in whale sharks. *Functional Ecology*. **25**, 595-607.
(10.1111/j.1365-2435.2010.01801.x)
18 Gleiss, A. C., Wright, S., Liebsch, N., Wilson, R. P., Norman, B. 2013 Contrasting diel patterns in
vertical movement and locomotor activity of whale sharks at Ningaloo Reef. *Marine Biology*. **160**,
2981-2992. (10.1007/s00227-013-2288-3)

- Martin, R. A., Hammerschlag, N. 2012 Marine predator–prey contests: Ambush and speed versus vigilance and agility. *Marine Biology Research*. **8**, 90-94. (10.1080/17451000.2011.614255)
- Whitney, N. M., White, C. F., Gleiss, A. C., Schwieterman, G. D., Anderson, P., Hueter, R. E., Skomal, G. B. 2016 A novel method for determining post-release mortality, behavior, and recovery period using acceleration data loggers. *Fisheries Research*. **183**, 210-221. (<http://dx.doi.org/10.1016/j.fishres.2016.06.003>)
- Sakamoto, K. Q., Sato, K., Ishizuka, M., Watanuki, Y., Takahashi, A., Daunt, F., Wanless, S. 2009 Can ethograms be automatically generated using body acceleration data from free-ranging birds. *PLoS one*. **4**, e5379. (10.1371/journal.pone.0005379)
- Andrzejaczek, S., Gleiss, A. C., Pattiaratchi, C. B., Meekan, M. G. 2018 First insights into the fine-scale movements of the sandbar shark, *Carcharhinus plumbeus*. *Frontiers in Marine Science*. **5**, (10.3389/fmars.2018.00483)
- Pinheiro, J., Bates, D., DebRoy, S., Sarkar, D., Heisterkamp, S., Van Willigen, B., Maintainer, R. 2017 Package 'nlme'. *Linear and Nonlinear Mixed Effects Models*, version. 3-1.
- R Core Team. R: A Language and Environment for Statistical Computing. In: R. F. f. S. Computing, ed. Vienna, Austria 2017.
- Zuur, A., Ieno, E. N., Walker, N., Saveliev, A. A., Smith, G. M. 2009 *Mixed effects models and extensions in ecology with R*. Springer Science & Business Media.
- Dowd, W. W., Brill, R. W., Bushnell, P. G., Musick, J. A. 2006 Standard and routine metabolic rates of juvenile sandbar sharks (*Carcharhinus plumbeus*), including the effects of body mass and acute temperature change. *Fishery Bulletin*. **104**, (<http://escholarship.org/uc/item/9g59202m>)
- Gleiss, A. C., Potvin, J., Goldbogen, J. A. 2017 Physical trade-offs shape the evolution of buoyancy control in sharks. *Proceedings of the Royal Society B: Biological Sciences*. **284**,
- Adachi, T., Maresh, J. L., Robinson, P. W., Peterson, S. H., Costa, D. P., Naito, Y., Watanabe, Y. Y., Takahashi, A. 2014 The foraging benefits of being fat in a highly migratory marine mammal. *Proceedings of the Royal Society B: Biological Sciences*. **281**,
- Vaudo, J. J., Wetherbee, B. M., Harvey, G., Nemeth, R. S., Aming, C., Burnie, N., Howey-Jordan, L. A., Shivji, M. S. 2014 Intraspecific variation in vertical habitat use by tiger sharks (*Galeocerdo cuvier*) in the western North Atlantic. *Ecology and evolution*. **4**, 1768-1786. (10.1002/ece3.1053)
- Ferreira, L. C., Thums, M., Meeuwig, J. J., Vianna, G. M. S., Stevens, J., McAuley, R., Meekan, M. G. 2015 Crossing latitudes—long-distance tracking of an apex predator. *PLOS ONE*. **10**, e0116916. (<https://doi.org/10.1371/journal.pone.0116916>)
- Bozzano, A., Collin, S. P. 2000 Retinal ganglion cell topography in elasmobranchs. *Brain, Behavior and Evolution*. **55**, 191-208. (<https://doi.org/10.1159/000006652>)
- Wilson, R. P., Quintana, F., Hobson, V. J. 2012 Construction of energy landscapes can clarify the movement and distribution of foraging animals. *Proceedings of the Royal Society B: Biological Sciences*. **279**, 975. (10.1098/rspb.2011.1544)
- Shepard, E. L. C., Wilson, R. P., Rees, W. G., Grundy, E., Lambertucci, S. A., Vosper, S. B. 2013 Energy landscapes shape animal movement. *The American Naturalist*. **182**, 298-312. ([doi.org/10.1086/671257](http://dx.doi.org/10.1086/671257))
- Adachi, T., Costa, D. P., Robinson, P. W., Peterson, S. H., Yamamichi, M., Naito, Y., Takahashi, A. 2017 Searching for prey in a three-dimensional environment: hierarchical movements enhance foraging success in northern elephant seals. *Functional Ecology*. **31**, 361-369. (10.1111/1365-2435.12686)
- Heithaus, M. R., Dill, L. M., Marshall, G. J., Buhleier, B. 2002 Habitat use and foraging behavior of tiger sharks (*Galeocerdo cuvier*) in a seagrass ecosystem. *Mar Biol*. **140**, (10.1007/s00227-001-0711-7)
- Meyer, C., Papastamatiou, Y., Holland, K. 2010 A multiple instrument approach to quantifying the movement patterns and habitat use of tiger (*Galeocerdo cuvier*) and Galapagos sharks (*Carcharhinus*

*galapagensis*) at French Frigate Shoals, Hawaii. *Marine Biology*. **157**, 1857-1868. (10.1007/s00227-
010-1457-x)
37 Heithaus, M. R., Wirsing, A. J., Dill, L. M. 2012 The ecological importance of intact top-predator
populations: a synthesis of 15 years of research in a seagrass ecosystem. *Marine and Freshwater*
*Research*. **63**, 1039-1050. (<http://dx.doi.org/10.1071/MF12024>)

Table:

Table 1. Results of generalised linear mixed models testing the relationship between seabed depth
 (MaxD) and vertical movement behaviours. All models were compared with null models using Akaike's
 Information Criterion (AIC) and conditional (R^2_c) and marginal (R^2_m) R^2 values. All models were run
 using the nlme package in R with shark identity included as a random effect. All null models **included**
 the random effect. Inshore indicates windows where maximum depth was <25 m, and offshore where
 maximum depth was >25 m.

Inshore Model	DF	AIC	R^2_m	R^2_c
Diving Ratio ~Maximum Depth	629	-831	0.35	0.55
Diving Ratio ~ 1	629	-629	0	0.42
Descent Pitch ~Maximum Depth	629	2724	0.35	0.48
Descent Pitch ~ 1	629	2903	0	0.31
Ascent Pitch ~Maximum Depth	629	2672	0.18	0.24
Ascent Pitch ~ 1	629	2743	0	0.14
Descent Vertical Velocity ~Maximum Depth	629	-2695	0.26	0.53
Descent Vertical Velocity ~ 1	629	-2555	0	0.30
Ascent Vertical Velocity ~Maximum Depth	629	2748	0.2	0.36
Ascent Vertical Velocity ~ 1	629	2668	0	0.18
Offshore Model	DF	AIC	R^2_m	R^2_c
Diving Ratio ~Maximum Depth	173	-165	0.09	0.39
Diving Ratio ~ 1	173	-168	0	0.37
Descent Pitch ~Maximum Depth	173	1062	0.002	0.26
Descent Pitch ~ 1	173	1054	0	0.27
Ascent Pitch ~Maximum Depth	173	919	0.007	0.15
Ascent Pitch ~ 1	173	912	0	0.11
Descent Vertical Velocity ~Maximum Depth	173	-348	0.03	0.26
Descent Vertical Velocity ~ 1	173	-360	0	0.35
Ascent Vertical Velocity ~Maximum Depth	173	-466	0.004	0.21
Ascent Vertical Velocity ~ 1	173	-482	0	0.23

Figure legends:

Figure 1. Schematic representation of an oscillatory movement, and the distance parameters used in
calculating the cost of transport.

Figure 2. Mean instantaneous ODBA for all 21 tiger sharks at a range of absolute (*a*) ascent and (*b*)
descent angles (placed in 5° bins). All sharks displayed a positive slope for ODBA ~ ascent angle. No
consistent relationship was found between ODBA and descent angle.

Figure 3. Gliding in tiger sharks over the course of (*a*) an entire track (10 hours); (*b*) 45 minutes; and
(*c*) 6 minutes. Glides are marked in red on the x axis. Dynamic angular velocity was taken from the
lateral (sway) axes recorded by the gyroscope. (*d*) The proportion of descent time spent gliding by
each individual tiger shark.

Figure 4. Relationship between dive angle and habitat depth. (*a* & *b*) Boxplots displaying differences
in ascent and descent pitch angle between habitat zones. Inshore denotes areas where maximum
depth within a window was <25 m, and offshore where depths were >25 m. Shark silhouettes angles
are not to scale. (*c* & *d*) Relationship between maximum depth in a sampling window and descent and
ascent angle in inshore zones (<25 m).

Figure 5. (*a*, *b*) Schematic diagrams representing how diving ratio may differ as a result of seabed
depth and sampling window. Depth traces have the same vertical velocity, and same time spent level
swimming at the surface and seabed in between vertical movements. (*b*) represents a habitat of twice
the depth of habitat (*a*). Red dashed square represents the fixed time window within which diving
ratio is measured. (*c*) Relationship between seabed depth (m) and diving ratio measured in 15 minute
windows. Maximum depth is a proxy for seabed depth. The red line displays diving ratio calculated if
time level swimming at the surface and seabed for oscillations remained the same at all maximum
depths.

Figure 6. The cost of transport of a single oscillatory movement of a tiger shark on (*a*) a horizontal
plane and (*b*) a vertical plane. Arrows indicate where the cost of transport is minimised in both cases.
Lines represent the cost of a given descent pitch angle (label on figure) in relation to varying ascent
pitch (x-axis).

Figure 1. Schematic representation of an oscillatory movement, and the distance parameters used in calculating the cost of transport.

160x80mm (300 x 300 DPI)

Figure 2. Mean instantaneous ODBA for all 21 tiger sharks at a range of absolute (a) ascent and (b) descent angles (placed in 5° bins). All sharks displayed a positive slope for ODBA ~ ascent angle. No consistent relationship was found between ODBA and descent angle.

109x119mm (300 x 300 DPI)

Figure 3. Gliding in tiger sharks over the course of (a) an entire track (10 hours); (b) 45 minutes; and (c) 6
minutes. Glides are marked in red on the x axis. Dynamic angular velocity was taken from the lateral (sway)
axes recorded by the gyroscope. (d) The proportion of descent time spent gliding by each individual tiger
shark.

129x245mm (300 x 300 DPI)

Figure 4. Relationship between dive angle and habitat depth. (a & b) Boxplots displaying differences in ascent and descent pitch angle between habitat zones. Inshore denotes areas where maximum depth within a window was <25 m, and offshore where depths were >25 m. Shark silhouettes angles are not to scale. (c & d) Relationship between maximum depth in a sampling window and descent and ascent angle in inshore zones (<25 m).

Can you also show the relationship for depths greater than 25m? I'm not clear on why you decided to bin depth. It looks like you have more than enough data to model depth as a continuous variable using something like a GAM to capture the non-linearity. Can you explain your rationale for your approach?

The language used in this caption is quite poor, the sentences are not grammatically correct. Can you improve it for clarity?

For example, why is maximum dive depth a proxy for seabed depth? It clearly isn't in example (a).

Figure 5. (a, b) Schematic diagrams representing how diving ratio may differ as a result of seabed depth and sampling window. Depth traces have the same vertical velocity, and same time spent level swimming at the surface and seabed in between vertical movements. (b) represents a habitat of twice the depth of habitat (a). Red dashed square represents the fixed time window within which diving ratio is measured. (c) Relationship between seabed depth (m) and diving ratio measured in 15 minute windows. Maximum depth is a proxy for seabed depth. The red line displays diving ratio calculated if time level swimming at the surface and seabed for oscillations remained the same at all maximum depths. What do you mean by "time level swimming"?

184x129mm (300 x 300 DPI)

It's not clear how this figure shows optimality. Can you add enough detail for this figure and caption to be able to stand alone and still be understood by a non-expert reader?

Figure 6. The cost of transport of a single oscillatory movement of a tiger shark on (a) a horizontal plane and (b) a vertical plane. Arrows indicate where the cost of transport is minimised in both cases. Lines represent the cost of a given descent pitch angle (label on figure) in relation to varying ascent pitch (x-axis).

180x80mm (300 x 300 DPI)

Appendix B

Response to referees

Depth dependent dive kinematics suggest cost-efficient foraging strategies by tiger sharks

Note that line numbers refer to those in the track-changed document.

Reviewer 1:

Comments to the Author(s)

This manuscript explained how tiger sharks change depth movement according to the depth of the sea. The model looks a little rough (The estimated optimal pitch angle does not appear to match the actual shark pitch angle), but I think it's worth publishing because the results are not so strange (I think it's interesting that the behavior changes depending on the habitat.). There were some wondering parts, but it was good because the explanation came out later.

Specific comments

L89 Although it is written to refer to Andrzejaczek et al. 2019, it is better to write the model and specifications (size, weight etc.) of the devices.

More detail about these devices and their specifications have been added to the main text at line 103-105. This reads: 'Either a CATS (Customized Animal Tracking Solutions, Australia) Diary Tag (dimensions and weight with clamp: 15 x 4 x 6 cm and 300 g) or CATS Cam Tag (23 x 4 x 7 cm and 500 g) were then clamped to the dorsal fins for periods of 7-48 hours (see table S1).'

L98 Although it is written to refer to Andrzejaczek et al. 2019, the definition of vertical phase of movements (how to distinguish descent, ascent and horizontal swimming. From pitch? change of depth?) should be stated here.

We acknowledge that this was an important step of our methods that should have been described in more detail. Vertical phases of movement were calculated using vertical velocity (change in depth over time). We have added an extra paragraph to describe this approach at lines 129-137.

This section reads:

'Vertical velocity (VV), defined as the rate of change in depth over a one second period, was used to split the depth record into vertical swimming phases ("ascending", "descending" and "level swimming"). This was executed by smoothing the depth record using a 10 s running mean and calculating the average VV by taking the difference of this smoothed depth between successive points at 1 s intervals. Ascents and descents were defined where VV exceeded an absolute value of 0.05 m/s for more than 10 s, and level where this value was not exceeded [14, 23]. As error in the depth sensor was minimal (± 10 cm), we do not believe sensor accuracy significantly affected vertical movement phase classification.'

L100 It is written as the pitch angle of vertical movement, but isn't it a shark angle? Is the shark moving parallel to the body axis (no angle of attack)?

This has been revised to 'shark body pitch angles (orientation of the shark with regard to the horizontal plane)' at line 117-118.

L119 Is there a tailbeat frequency even during gliding? Is it okay to extract gliding with the same "mask" for all individuals, even though there are slight differences in the attaching position and differences in individual sizes?

A low tailbeat frequency value was still often recorded during gliding behaviour as a very small signal was still recorded by the tri-axial sensors (as you can see by the very small signal in sway angular velocity in the figure below). Gliding was distinguished more by very low values of angular velocity signal amplitude (line 147-148).

Threshold values for the mask analysis were calculated separately for each individual to take into account difference in shark size and tag attachment positions. We have now made this clearer at lines 151-155 of this section.

L122 Although the ascent and descent speeds were calculated only when the pitch angle was 20° or more, the other analyses that appear later have a pitch angle smaller than 20° (e.g. Fig. 4). The validity when the pitch angle is shallower than 20° should be shown.

The referee is correct that our method for estimating swimming speeds (and therefore the input of our model) relies on steeper angles and thus we had to make the assumption that the constant speeds we have measured at angles $>20^\circ$ also applies to shallow angles $<20^\circ$. Our ODBA vs pitch angle data supports the validity of the assumption, because the patterns follow theoretical expectations (no patterns in the relationship between descent angle and ODBA and an exponential relationship between ODBA and ascent pitch) with no observed break in the relationship at shallow angles. If speeds at shallow angles were significantly different, one would expect ODBA to be significantly larger (if speeds were higher) or lower. This is especially true as changes in swimming speed have a greater effect on energetics than pitch angle (Nakamura et al 2011-MEPS; Papastamatiou et al 2018 – Sci Reports). Since no such break was observed, we are confident that no systematic change in swimming speeds occur at shallow angles.

L128 Again, it is necessary to define vertical and horizontal swimming.

Vertical and horizontal swimming have now been defined in lines 129-137 as outlined above.

L172 It is necessary to state what formula you used to regress. Although it is written as a quadratic curve in the result, is it part of a sine curve due to the relationship between gravity and angle?

The best fitting curve displayed a quadratic relationship between ODBA and ascent angle with the equation $ODBA = 2E-05 \varphi_a^2 + 0.00001\varphi_a + 0.0226$. We have added this equation to the methods at line 247-248, and the caption for Figure 1. The mechanisms behind the curve is likely related to the efficiency of lift production shifting, as a result of the increasing angle of attack the counteracting gravity will shift from the pectoral fins to the caudal fin. So yes, the relationship is likely sinusoidal, however, since there is currently nothing published on the mechanisms associated with it, we feel that a data-driven approach is more compelling.

L179 No relationship between pitch angle and speed is declared here. Although it is written in the result, I think that it is better to quote the table S2 here and explain the reason.

We agree, and have moved this reasoning and a reference to table s2 to this section of the methods (at lines 254-255), and have removed these details from the results.

L208 Are there any gliding ascent?

We only recorded very short, unsustained periods of gliding on ascent with no gliding on ascent recorded by half of the tagged sharks. We have added a sentence describing this at line 302-304. This reads 'In addition, only very short, unsustained (<3 seconds), periods of gliding were recorded on ascent, with 50% of sharks recording no gliding on ascent.'

L232 Is it possible to calculate the cost of level swimming along the seabed?

We were not able to reliably calculate the cost of level swimming along the seabed for two reasons.

1) It was not possible to know exactly when an individual was swimming along the seabed. Maximum depth was able to be used a proxy for seabed depth within a 15-minute time window, however, within shorter periods this would be less accurate.

2) Our methods for splitting the depth record into the vertical phases of locomotion meant that periods of ascent and descent were conservatively estimated (ascent and descent were defined as periods where VV exceeds certain values for > 10 seconds). So while ascent and descent periods would be accurately labelled, level swimming would be less so.

L272 It is better to cite this paper because it is a shark paper rather than a seal one. Del Raye et al. (2013) Travelling light: white sharks (Carcharodon carcharias) rely on body lipid stores to power ocean-basin scale migration. Proc R Soc B 280:20130836. And this sentence has no verb.

This citation has been added as suggested at line 382.

L286 It is better to examine the relationship between glide rate and pitch angle.

We agree with the reviewer. We have revised both the results (lines 306-307) and discussion (lines 410-411) here to explore how body pitch on descent changes when individuals are gliding versus when they are exhibiting active descents.

L304 Isn't it strange to use the same vertical velocity when estimating the dive ratio in Fig. 5 despite the result that the pitch angle changes between inshore and offshore?

As a result of comments and confusion surrounding this figure from both reviewers, and discussion with all co-authors, this figure has been removed. This did not lead to any significant changes in the results or discussion.

L314 Looking at Fig. 4, the pitch angle changes continuously as it gets deeper (especially ascent), so I think there should be a discussion about that. Is it possible to trade off the horizontal and vertical directions with a shallower pitch angle than the model that covers the vertical direction?

Increasing pitch angle with depth was discussed at lines 435-439.

We don't believe that there would be a tradeoff between horizontal and vertical directions with a shallow pitch angle. The expense modelled here is time, and at a shallower angle there would be smaller vertical distances covered per unit time.

Fig. 1 How about overlaying glides on a depth profile (e.g. colouring only glides)?

As Figure 1 is a schematic, we haven't overlaid glides here. Figure 3 displays gliding at a variety of scales, and we have now also marked glides on the finest scale plot (Fig 3c).

Fig. 4 The shark silhouette is steep even though the offshore ascent angle is smaller. In results, "Mean absolute pitch angles increased between inshore and offshore habitats from 9 ± 2.3 to 10.7 ± 3.2 for ascent." Has inshore and offshore been switched?

Apologies for this oversight. Inshore and offshore have indeed been switched. Fig. 4 has now been corrected.

Fig. S1 It is better to swap left and right so that it fits in Fig. 4.

Both Fig 4 and Fig S1 have been reformatted.

Reviewer 2:

Comments to the Author(s)

Dear Dr Andrzejczek,

I have read your manuscript, which is on the whole well-written and raises an interesting topic of oscillatory movements in tiger sharks, provides measurements, and makes inferences about energy efficiency of movement during these "dives", with respect to dive angles.

There are some aspects of your work that cause me to have concerns about the validity of the conclusions drawn from the data you present here. Perhaps I haven't understood correctly, but you claim that tiger sharks adopt optimal dive angles during oscillatory movements. I am uncomfortable with this claim because optimality is a very strong word, meaning something quite specific, which I don't think you have measured. For example, you have not mentioned or measured the potential profitability of the different environments that sharks encountered. I'm not suggesting that you ought to have measured that; in general it's hard to do and most of us don't have that information about the environment our study species travel to. However, I think it's essential to mention and discuss it, and very clearly define in what sense you consider movements optimal. In addition, you make a comparison between inshore and offshore habitats,

which is treated as a categorical variable. From figure 4, it appears that there was sufficient data to treat depth as a continuous variable, which would make better use of the resolution of the data and be more convincing.

The statistical modelling aspect of your work appears generally sound, but it's difficult to tell without more detail. You have not provided the code you used to carry out the modelling, so at the moment it's not reproducible even though you have provided the data. I would encourage you to always submit your code, in keeping with the ethos of reproducibility, without which a reader can't replicate your work. One question that lingers in my mind is why you have used a one-way ANOVA to compare the mechanical cost of ascending versus descending. Perhaps I've missed something, but the only reason to do a one-way ANOVA instead of the two-way ANOVA is if it was impossible to have an effect go in both directions (bigger and smaller). That doesn't seem to be the case here. Can you explain why you used a one-way ANOVA?

To summarise my main concerns with the paper:

1) the use of the word optimal is problematic, because it implies something very specific, and which I don't think you can back up, owing to the lack of information about how profitable a habitat is. My argument is that optimality implies a balance of input and output. In this manuscript you discuss measurements of output - energy spent - but you have no measurements of input - energy gained, which is the usual situation in studies of animal movement. My issue isn't that you lack these data, but rather that you don't discuss the issue of varying profitability. It may become worth spending extra energy and moving "sub-optimally" if the returns outweigh the costs. This isn't mentioned in your manuscript.

This is a very valid point, and we thank the reviewer for pointing this out. We have made sure to revise our manuscript to discuss our results in terms of the calculated angles that **conserve energy** while travelling and searching for prey, rather than **optimizing** movement or the cost of transport/movement.

We also now note that we did not measure energy inputs and discuss that these are also a key component of bioenergetics modelling at lines 481-492, and that deep environments may be more profitable if more energy can be gained from prey here despite sub-optimal movement costs.

2) one of the central outcomes of this work appears to be that tiger sharks dive at different angles in shallow and deep water, shallow sandflats being mentioned as important habitat where sharks can hunt while maintaining optimal angles. I find this conclusion problematic because most of the sharks in your study have length over 2.5m. This means that they physically cannot make descents or ascents past some maximum absolute angle. Perhaps I have misunderstood something here but again, the text needs to be made much clearer if that's the case.

We thank the reviewer for this comment. This is a very valid point, and one we did not originally investigate. Interestingly, we now looked at the maximum angles of ascent and descent with water depth, and found that though there is likely a physical limit to dive angle in shallow waters, this limit is relatively steep, and sharks are still capable of utilizing angles up to at least 25° here (see figure 1 below).

Figure S1. Relationship between diving angles and maximum depth (proxy for seabed depth). A and B display the full range of maximum angles on ascent and descent respectively with seabed depth, and C and D show the same relationship with the y-axis limited to a maximum of 25°.

However, we cannot discount the possibility that other mechanisms are driving these dive angles in shallow waters e.g. sharks are using the best angles to search for prey here. But this doesn't change the fact that, energetically (from a cost of transport standpoint), there are benefits to moving at these shallow angles. We have now included statements to this effect in the discussion at lines 439-456.

3) the methodology and analysis are not well documented. As far as the data preparation is concerned, I would have liked to see much more detail, perhaps in an appendix, about how the data were processed and prepared for analysis, and details about the tag measurements. For example, what was considered a dive? What was the measurement error of the instrument? In addition, I would have liked to see a more

comprehensive study of the effect of different time windows that could support the choice of time window in the analysis. Similarly, I would like to see a much more detailed rationale of the statistical modelling, including reasons for choosing the model you present, stating clearly the response variables and covariates, candidate models, error distributions and error structures, reason for choosing an autocorrelation function to model the errors and the way the final model was chosen, including a mention of the limitations of using AIC to do model selection. All of this could go in an appendix but it's critical that you explain it somewhere.

We apologize that our methods and analysis were not clear.

We have firstly expanded the detail in both the appendix and main text methods about how the data were processed and prepared for analysis (more detail in the specific comments below). We have provided extra detail in the main text about the devices used, their measurement error, and added a paragraph on how the depth record was split into ascents and descents (i.e. how dives were defined). In the supplementary information, we have added an expanded methods section, including information on how the tri-axial sensor data were processed and how the recovery period was calculated.

In our 'Data analysis' section we have added more information stating the rationale behind using mixed models and our autocorrelation structure (more detail in the specific comments below). We have also increased the clarity of the list of response variables and covariates modelled, and how model selection was performed. Using AIC is a robust approach to model selection with no clear limitations, however, we have also added the results of likelihood ratio tests to Table 1 to compare nested models (i.e. comparing each model with its null model). This revealed consistent results with models selected by AIC.

Lastly, while we agree that our time window may be somewhat dependent on the habitat depth of our study site, we have acknowledged throughout the dataset that we are looking at a coastal system. We have also added additional detail to our supplementary of how the 15-minute sampling window was chosen and caution that this could change in deeper habitats (see specific comments for more detail).

Specific comments from PDF (page and line numbers refer to those generated by the authors)

Page 0 Line 28: Well done for providing the data but without the code used for data preparation and statistical analyses it's not possible to reproduce your approach.

We have now provided a much more detailed methods section in the appendix. As data were prepared and processed in four different softwares (R, Microsoft Excel, Igor Pro and Framework 4), the majority of which rely on point-and-click workflows and not raw code, we were unable to provide a single source code that could be used in a practical manner to allow direct reproducibility.

The statistical analyses followed a standard framework and code for GLMMs available in Zuur et al. (2009). We therefore do not think it is necessary to add statistical code to the appendix.

Page 1 Line 20-21: This is confusing, I have made more comments below. Where they optimal even at 1degree? What does this mean biologically? Is there any uncertainty associated with this conclusion?

As mentioned above, we have added edits throughout the manuscript to discuss our results in terms of conserving energy and reducing the cost of transport, instead of 'optimising' energy expenditure.

Here, we have clarified the angles at which the cost of transport is reduced, and revised our wording in terms of optimality at lines 20-27. This section now reads:

‘The cost of horizontal transport was minimized by descending at the lowest possible angle and ascending at an angle of 5-14°, meaning that vertical oscillations conserved energy relative to swimming at a level depth. Reduction of vertical travel costs occurred at steeper angles. The absolute dive angles of tiger sharks increased between inshore and offshore zones, presumably to reduce the cost of transport while continuously hunting for prey in both benthic and surface habitats. Oscillatory movements of tiger sharks conform to strategies of cost-efficient foraging, and shallow inshore habitats appear to be an important habitat for both hunting prey and conserving energy while travelling. ‘

Page 3 Line 36: This isn't a good reference for general cost-efficiency of movement in top predators. The paper is about obligate scavengers.

Fair point, citation had been changed to: 9 Carbone, C., Teacher, A., Rowcliffe, J. M. 2007 The Costs of Carnivory. PLOS Biology at line 47 and 10 Gleiss, A. C., Jorgensen, S. J., Liebsch, N., Sala, J. E., Norman, B., Hays, G. C., Quintana, F., Grundy, E., Campagna, C., Trites, A. W. 2011 Convergent evolution in locomotory patterns of flying and swimming animals. Nature Communications. 2, 352. (10.1038/ncomms1350)

Page 3 Line 40: Does this still count as a "dive"? How do you scale the definition with available depth?

This is a great question, and the answer can change between the many and varied definitions of a ‘dive’ in the existing literature.

We do not have the data to directly assess this, but based off our calculations of vertical movement phases (ascents, descents and level swimming) where we use 10 seconds of continuous ascending or descending to label these sections (now more clearly described in lines 129-137), these movements in Heithaus et al (2002) would be considered a dive by our definition (bounces lasted an average of 4.3 minutes, with vertical phases of average 0.16 m/s and 0.10 m/s for descent and ascent respectively; Heithaus et al 2002).

Page 3 Line 48-49: Has area-restricted search behaviour been shown to be more strongly linked with foraging than directed movements? If not, there is no way of knowing whether hunting behaviour is linked to movement patterns.

ARS has been linked with increased foraging success in a number of marine animals (e.g. Austin et al. 2006; Papastamatiou et al. 2009; Adachi et al 2017), indicating that hunting behavior is indeed linked to movement patterns. A sentence to this effect has been added at line 59-60 to clarify this point.

Page 4 Line 67: This depends on the type of prey. Dead whales will float, for example.

This is a good point. Line has changed to ‘given that *most* dead prey will sink to the bottom’ at line 82.

Page 4 Line 76: I'm a bit confused here, do they search while changing depth? Or do they also hunt while changing depth? It's not clear that it would always be better to do this as fast as possible

This section relates to when they are searching/hunting the surface and seabed for food, and transiting the water column in between. We have added ‘between the surface and seabed’ at line 91 to increase the clarity here, so it now reads: ‘We therefore predict that the diving energetics and kinematics of these animals will be dependent on the depth of the habitat being occupied, with deeper waters featuring

steeper dive angles when searching for prey to allow for faster transit times between the surface and seabed.'

Page 4 Line 86: How many sharks were tagged? I know you have table in S1 but it would be good to state it here too

22 tiger sharks were captured and tagged. This has been added at line 101.

Page 5 Line 118-121: It's not clear what this is describing, what is a mask? Many of your readers might not know. I know you refer to an already published paper [22] but if you are going to describe the behavioural classification routine, even briefly, it needs to be clear.

We acknowledge that 'mask' is a confusing term, and have changed it to 'threshold values'. The section now reads '(2) threshold values of angular velocity signal amplitude and tailbeat frequency were set using the characteristics of correctly classified gliding behavior (from visual inspection of the dynamic sway data and concurrent videos). These thresholds were then used to extract glides from all sharks, and an additional manual quality control was undergone in the case where the mask obviously misclassified glides' at lines 151-155.

Page 6 Line 126-127: What is the uncertainty associated with depth? This will directly affect pitch angle.

Depth sensor accuracy was ± 10 cm (added at line 136). We do not believe that this small error level would have a significant effect on measures of pitch angle (now stated at line 135-137).

Page 6 Line 131-132: Does this make it highly dependent on this particular dataset?

The time period with highest variance in turning window varied slightly between individuals, but 15 minutes was the highest in most cases. Though this chosen time-window may be dependent on our study site, we have acknowledged that we are investigating tiger sharks in a coastal system throughout our manuscript. If similar methods were used where deeper habitats were present (e.g. Hawaii), sampling time windows may need to be extended. To facilitate this, we have added more detail to the supplementary methods, as well as a figure displaying how our chosen sampling window captured dives in their entirety in habitats of different depths (Figure S2), and point out that in deeper habitats than that encountered here, the sampling window would likely need to be larger. We have also added Figure S3 to the Supplementary as a schematic demonstrating how variance in turning angles was used to calculate the sampling window.

Figure S2. Depth time-series for three different individuals moving in three different depth zones. Grey and white bars indicate 15-minute sampling windows and how they cover dives in each different zone.

Figure S3. Schematic diagram of how variance in turning angle was investigated among individual tiger sharks. Each coloured line represents an individual shark. Dashed line at 900 seconds (15 minutes) demonstrates overlap with highest number of individuals.

*Page 6 Line 137: Did you allow for any kind of random variability or error around a depth measurement or was *any* change in depth considered as “moving vertically”? Does a tiger shark ever maintain depth without any variability in depth? What was the accuracy of the depth sensor?*

As now described in lines 129-137, the depth record was split into vertical moving (i.e. ascending and descending) and level swimming phases by smoothing the depth record using a 10 s running mean and calculating the average VV by taking the difference of this smoothed depth between successive points at 1 s intervals. Ascents and descents were defined where VV exceeded an absolute value of 0.05 m/s for more than 10 s, and level where this value was not exceeded.

Tiger sharks often swam at a level depth, moving vertically for a mean of $38.4 \pm 26\%$ of the track (line 293).

Depth sensor accuracy was $<\pm 10$ cm, and we do not believe this small level of error would significantly affect categorization of vertical movement phases (added at line 135-137).

Page 6 Line 141-143: Why do you need a mixed model for this? Did you just have a random intercept or also random slopes? It would be good to add a sentence explaining the rationale.

A mixed model with a random intercept was used to control problems associated with non-independence due to repeat measurements of behaviour from the same individual (Zuur et al 2009; Harrison et al 2018).

We have now added the following sentence at line 189-191: ‘The use of mixed models allowed us to set individual tiger shark as a random intercept effect and therefore control problems associated with non-independence due to repeat measurements of behaviour from the same individual (Harrison et al 2018).’

Page 6 Line 144: Was this the only explanatory variable?

Maximum depth was the only explanatory variable, as outlined in Table 1, and line 193. We have made this cleared by re-arranging line 191-193 which now reads: ‘The maximum depth (m) recorded within each time window was used as a proxy for seabed depth (based on video analysis; see Andrzejaczek et al. [14]) and was *set as the only* fixed explanatory variable for all models.’

Page 6 Line 146: What kind of temporal autocorrelation does this function assume? Did you look for what kind of autocorrelation was present in the data? What are the consequences of mis-specifying the shape of the autocorrelation?

As time-series data is inherently auto-correlated and often violates assumptions of independence of samples, we modelled the serial dependence of our data and accounted for it using the corAR1 function in R. We have increased the detail in this section outlining our reason for using this method at lines 194-200. This added section reads:

‘As time-series data is inherently auto-correlated, we modelled the serial dependence in our data using an auto-regressive process of order 1 (AR1), which assumes that the magnitude of the data at time t is affected by the magnitude of the data at time $t - 1$ [21, 29]. Auto-correlation was tested on initial model fits without a correlation structure, revealing a steady decline of serial correlation with increasing lag from time t . The correlation at lag = 1 was then used in specifying the correlation structure of the data [29] and added as a final term to each model using the corAR1 function in R.’

Page 6 Line 148: Windows in what? This comes across as vague here

15-minute sampling windows. This has now been added at line 201-202 so that it reads ‘Together with nautical charts from Ningaloo Reef, maximum depth was used to classify 15-minute sampling windows as either ‘inshore’ (<25 m, inside the reef) or ‘offshore’ (>25 m, outside the reef).’

Page 6 Line 150: Do you mean unequal variance in the residuals with depth? Heterogeneity is perhaps harder to understand, you could add it in brackets. Why is it bad to have an unbalanced design? I think it’s worth mentioning bias here. Both of these statements could use a little bit of context. One or two sentences would really help a lot.

We have specified what we mean by these terms and added some context to this section as suggested. It now reads ‘Initial model runs considered the entire dataset, however, only four tiger sharks entered offshore habitats resulting in an unbalanced model design and therefore biasing predictions in deeper water to these individuals. In addition, residuals displayed unequal variance with depth (i.e. were heterogeneous). To overcome these issues, inshore and offshore periods were considered in separate GLMMs’ at lines 203-207.

Page 6 Line 151: What were your candidate models? How did you chose them?

All candidate models are listed in Table 1. Only two candidate models were ran for each response variable (a) the null model and b) with maximum depth as an explanatory variable), and subsequent selection took place using AIC ranks (the model with the lowest AIC being selected).

We have increased clarity throughout this section in terms of the above, now refer to Table 1, and have added a line (at line 207-208) highlighting that the model with the lowest AIC was selected following AIC ranking.

Page 7 Line 154-160: This section is confusing, again, I think some context would help.

As a result of comments and confusion surrounding this figure from both reviewers, and discussion with all co-authors, this figure has been removed. This did not lead to any significant changes in the results or discussion.

Page 7 Line 159: Where do the four 15 min windows come from?

Section has been removed (as described above).

Page 7 Line 167-168: The average ODBA for descent and ascent? What qualified as a descent and ascent? You define these below but it would be good to bring them up.

As above, the categorization of ascent and descent have now been defined in the added section at lines 129-137.

Page 7 Line 171: How did this compare to values from the literature for tiger sharks or other species of shark?

Unfortunately, we are unable to compare this value with others in the literature as this value is dependent on the location of tag attachment on the shark (Whitney et al 2012 - Integrative multi-sensor tagging of elasmobranchs: emerging techniques to quantify behavior, physiology, and ecology. In Biology of sharks and their relatives). While the first dorsal fin is the only one large enough to carry the tag package in tiger sharks and is sufficient for detecting tailbeat activity and pitch angles, the second dorsal fin (the location

of attachment for whale sharks in Gleiss et al 2011) can reveal more subtle changes in tailbeat activity. Either are feasible for quantifying energy expenditure, but more posterior-located loggers will experience greater acceleration amplitude and therefore have larger ODBA values than more anterior-placed units (see figure from Whitney et al 2012 below). We therefore cannot compare between studies with different tag attachment locations, especially since the position of the dorsal fin differs between different species of sharks. Standardisation of attachment location within a study, as is the case with our tiger sharks, is paramount.

FIGURE 9.9

Swaying (lateral) dynamic acceleration (A, C) and overall dynamic body acceleration (B, D) traces of a captive blacktip shark (*Carcharinus limbatus*) tagged with two accelerometers, one located on the first dorsal fin (gray line) and the other attached to the second dorsal fin (black line). Note the overall larger variability present in the sway (C) and the larger ODBA (D) recorded with loggers attached to the second dorsal fin.

Page 7 Line 179-180: Why did you bin the pitch angle?

Apologies, we did not bin pitch angle, we instead calculated the total cost of ascents and descents at 5° increments of each. The text in this section has been changed at lines 259-261 to reflect this.

Page 8 Line 193: Did you have any sharks moving at or very close to the surface? Is there higher turbulence at the surface which might increase cost of transport? You talk about this below, is it relevant to mention it here too?

We thank the reviewer for pointing this out. Yes, it is relevant to mention here as the surface two meters was removed from the analysis of ODBA when ascending/descending to prevent superficially high ODBA levels being included. We have now added this point to line 244-247 so it now reads: ‘Ta, Td, ODBAa, and ODBAd, were all calculated depending on pitch angle (ϕ), with ODBA estimates excluding those taken from the top 2 m due to the effects of wave action at the surface frequently creating superficially high ODBA levels here (confirmed by video data)’ and cited Watanabe et al 2019-JEB, as a similar phenomenon was recorded here by white sharks.

Page 8 Line 196: It would still be useful to know how many tags were deployed to begin with.

The number of tiger sharks initially caught and tagged has now been added at line 101 (n =22 tiger sharks).

Page 8 Line 203: What is the measure of error you give here? $11.6-17.5 = -5.9$ which would be above the surface

Results are reported as mean \pm standard deviation as now indicated at line 291.

We thank the reviewer for pointing out these values. Mean depth was originally calculated as the mean of the entire dataset. We have now recalculated this value as the grand mean i.e. the mean and standard deviation of each individual's mean depth and replaced the value at line 291. This also corrects for biases toward individuals that had longer tag deployments. The standard deviation is also now smaller than the mean, so tiger sharks would not be estimated to be above the surface.

Page 8 Line 205-206: I would argue that this is the main result of your study. There is a lot of work that has gone into dive angles and body density in air-breathing divers, e.g. Aoki et al. 2011 Journal of Experimental Biology. This aspect of your work is a generally interesting result, beyond tiger sharks in your study area. Sorry to sound negative but the claims about energy expenditure inshore and offshore as a function of dive angle are less convincing, the way they are currently presented.

We thank the reviewer for this comment. These differences in dive angles are one of the main results of our study, being a 'dive kinematic' as introduced in the title. Following edits from other comments, and the additional methods presented following revision, we hope that the reviewer finds our other results more convincing.

Body density may be driving differences in gliding behaviours and dive angles between individuals and we thank the reviewer for pointing out Aoki et al 2011. It would be very interesting to investigate such relationships with body density, but unfortunately, we do not have drag coefficients and/or body mass estimates available to enable us to calculate this parameter. Hopefully, drag coefficients may soon become available to us through photogrammetry initiatives such as 'Digital Life' (<http://digitallife3d.org/>), where accurate 3D models of sharks are being created.

Page 9 Line 213-214: Can you say something about body density from these glides?

This is an interesting question, however, as noted in the comment above, we currently do not have body mass or drag coefficients for these animals and therefore cannot make inferences about body density here, other than that it is greater than that of the water the sharks are swimming in. This could definitely be an area for future focus once such estimates are obtained.

What we can say is that these animals are negatively buoyant (line 387-388), and that the individual differences in gliding behaviour may indeed be influenced by body density. Interestingly, we looked at differences in gliding behaviour between individuals in more detail and found that gliding occurred at steeper angles than active descents ($-14.9 \pm 5.7^\circ$ versus $-11.11 \pm 3.6^\circ$), and differed substantially among individuals (see figure below, also added to the supplementary information). These findings have been added to line 304-307 and line 381-382 of the results and discussion respectively, and Aoki et al 2011 has been added as a citation at line 383.

Figure S4. Individual differences in absolute angles on descent for A) all descent data, and B) passive gliding descents only. Note that in B), TS3 and TS20 are not present due to no gliding behaviour being exhibited by these two individuals. The red dashed line indicates the mean descent angle (11.1°). Also note that TS17 and TS24 are the same individual, tagged 11 days apart).

Page 9 Line 216: Do you mean biologically significant, or statistically significant? If the latter, at what level?

We mean statistically significant, and have now clarified this at line 316. As we performed model selection using AIC, we cannot report the level of significance here. Instead, we now also performed likelihood ratio

tests between the model and its nested null model, and presented these results in table 1. For all models selected over the null models, this significance was $p < 0.0001$.

Page 9 Line 217: GLMMs: Is there more than one inshore and one offshore model? How come? It would be good to say how many models you fit and how they were structured

We have increased the clarity of the model structures in the methods section (Generalised linear mixed models at lines 189-193 and 203-207) and listed each individual model tested in table 1. Only two models were tested per response variable in each habitat type, and the retained/selected model has now been bolded in table 1. In addition, we have now again listed each of these response variables tested on lines 318-319 of this results section.

Page 9 Line 231: What made you chose a one-way ANOVA? Was it not possible for the cost of ascent to be smaller than that of descent? In general, it's always better to run a two-way ANOVA to allow for a fair comparison, unless it's really impossible for an effect to go in both directions. That doesn't seem to be the case here.

One-way ANOVA tests the equality of means between two or more samples and is not directional. Two-way ANOVA examines the influence of two different categorical independent variables on one continuous dependent variable. In this case, the one-way ANOVA is the appropriate test.

Page 10 Line 249-250: I don't think you are able to make this claim. You haven't measured the profitability of different environments so you can only say there was a higher energy expenditure in certain habitats, but the net gain could have been much higher, if the profitability was much higher too. I also find it quite a confusing statement. Maybe I've missed the point, but it comes across like you are saying that they optimise diving behaviour when they are diving, since they can only have an absolute swim angle greater than zero when they are diving. Can you make it clearer?

We have clarified and revised this sentence so it now reads 'We found that the cost of transport was reduced on both horizontal and vertical planes at absolute ascent and descent angles greater than 0° (figure 6A), meaning that vertical oscillations conserved energy compared to swimming at a level depth' at line 346-361. Notably, we replaced 'the cost of travel was *optimized*' to the 'cost of transport was *reduced*'.

We thank the reviewer for addressing the point that we haven't incorporated other factors of the bioenergetics model, such as energy gain/input, and therefore cannot make claims about optimizing the cost of transport. We have now raised this point in the 'Seascapes effects on energy expenditure and gain' section of the discussion at lines 480-493. This section now reads: 'Very shallow environments (<3 m), such as the sandflats of Ningaloo Reef, represent a landscape where tiger sharks can move and forage at the most cost-efficient dive angles. However, while we were able to estimate energy output in the form of activity levels, we were unable to measure energy input, or more specifically, food assimilation, a key component of bioenergetics modelling and therefore estimating habitat profitability [38]. If deeper environments have higher prey densities, than the inputs in such habitats may outweigh the costs of moving sub-optimally, and ultimately may be more profitable. Although the data is not available to model food availability in this system, qualitative data from pseudo-tracks, video data, and the ecotourism industry have highlighted the importance of the shallow sandflat environments at Ningaloo Reef for foraging tiger sharks [14].'

Page 10 Line 258: This is a much better, less provocative way of framing your results. Optimality is a thorny issue that will always raise eyebrows.

We agree and have altered the discussion to frame our results in this way.

Page 10 Line 259-261: This is confusing, I can't tell what you mean here

We agree that this sentence is confusing, and have revised this and the next sentence to increase clarity. It now reads 'Our empirical models of the cost of transport showed that shallow angles of ascent and descent allowed tiger sharks to reduce their cost of transport in the horizontal plane relative to swimming at a level depth, whereas steeper angles reduced their energetic costs in the vertical plane' at lines 370-372.

Page 11 Line 279: This is still a confusing way of putting it, please consider revising

We agree, and have revised this sentence to read 'When ascents and descents were considered together and compared to swimming an equivalent horizontal distance at a level depth, the cost of horizontal transport was found to be minimized through the former, two-stage mode of oscillatory swimming' at lines 389-391.

Page 11 Line 299: I'm afraid this comes across as an obvious statement. If they are heading for the seabed then it follows they need to spend more time "diving"

We have decided to retain this sentence as, while obvious, we believe it is important to describe this pattern. If tiger sharks were continuously oscillating, and not searching the surface and seabed for prey in between vertical movements, then the diving ratio would remain constant.

Page 11 Line 301-302: Does this bring into question the value of your results?

However, as confusion came from both reviewers over this analysis, the figure, and the discussion of its results, and the fact that the change in diving ratio was already discussed in this context (at Lines 426-428), we decided to remove this analysis from our manuscript. This did not lead to any significant changes in the results or discussion.

Page 13 Line 341-344: This a tricky point because the habitat essentially enforces shallow dive angles, so to what extent can you say this is a strategy? In addition, how long is an average tiger shark? If they are in the range of 2-3 metres in length then there is a hard limit to the angle they can form

We thank the reviewer for this comment. As stated above, this is a very valid point, and one we did not originally investigate. Interestingly, we now looked at the maximum angles of ascent and descent with water depth, and found that though there is likely a physical limit to dive angle in shallow waters, this limit is relatively high, and sharks are still capable of utilizing angles up to at least 25° here (see figure S1 above).

There are also other mechanisms that may be driving these dive angles, and energy savings may be an added benefit, rather than the strategy itself. For example, these angles may be those best used to view the surface and seabed while searching for prey. We now state this at Line 440-456 of the discussion.

Page 13 Line 356: What do you mean here? Can you justify and explain this claim?

As Ningaloo Reef also has significant turtle and dugong populations (both of which are important grazers in seagrass and coral reef ecosystems), we believe it is likely that tiger sharks also play an important role in influencing the movements and behaviours of these species at this location. We have made this cleared by adding a statement to this effect at line 502-503.

Page 13 Line 359: I feel like this is an overly bold claim. The balance of energy is made up of both inputs and outputs. You have only measured outputs. What if deeper areas are much more profitable than sandflats? You don't mention this at all.

As noted above, we have now added sentences to the discussion (at lines 482-493) detailing our lack of measurement of inputs, and how this may affect optimization of the bioenergetics model. We have also now edited and softened this line of the conclusion so that it reads 'Collectively, our data showed that the oscillatory movements of coastal tiger sharks are the product of a combined strategy to search for prey while reducing the cost of transport' at line 506-507.

Tables and figures:

Table 1 (Page 18 Line 509): s deleted from included

Done.

Figure 2: How did you fit this regression line?

The line was fitted with a quadratic relationship. This has now been clarified in the methods at line 247-248, the results at line 334, and the equation placed in the figure 2 caption.

Figure 3: What does cycle mean here?

Cycle on the y-axis of this figure represents the inverse of tailbeat frequency (i.e. the length of time it takes to complete one tailbeat). We have now clarified this in the figure 3 caption.

Figure 4: Can you also show the relationship for depths greater than 25m? I'm not clear on why you decided to bin depth. It looks like you have more than enough data to model depth as a continuous variable using something like a GAM to capture the non-linearity. Can you explain your rationale for your approach?

As discussed above, the full relationship is displayed in the supplementary information at Fig S4. Depth was analysed as a continuous variable, but models were ran separately for inshore and offshore data. This was as only four sharks entered offshore waters, and the data if run as one model, results in an unbalanced model with heteroscedastic residuals (even with a GAMM).

Figure 5: The language used in this caption is quite poor, the sentences are not grammatically correct. Can you improve it for clarity? For example, why is maximum dive depth a proxy for seabed depth? It clearly isn't in example (a).

Can you clarify the caption for (b)?

What do you mean by "time level swimming"?

As discussed above, this analysis, figure and related results have now been removed.

Figure 6: It's not clear how this figure shows optimality. Can you add enough detail for this figure and caption to be able to stand alone and still be understood by a non-expert reader?

We agree that this figure caption was not clear. We have rewritten it to read: 'Figure 5. The horizontal cost of transport (a) and the vertical cost of transport (b) for a tiger shark executing a single oscillatory movement from the surface to 10 m according to descent and ascent pitch. Lines represent the cost of a given descent pitch angle (label on figure) in relation to varying ascent pitch (x-axis). Arrows indicate where the cost of transport is minimised in both cases. Reduction of the cost of transport in the horizontal (COT_{HD}) favours descending at minimal pitch during descent and ascending at an angle of 5-14°. Reduction of vertical travel costs (COT_{VD}) occurred by ascending and descending at steeper angles, although potential energy savings at absolute ascent and descent angles steeper than 20 and 15° respectively are minimal.'

Appendix C

Response to referees

Depth dependent dive kinematics suggest cost-efficient foraging strategies by tiger sharks

Associate editor comments:

Comments to the Author(s):

Thank-you for re-submitting your manuscript. Both reviewers and I agree that you have addressed their previous comments; the manuscript is now nearly ready for publication. However, I am returning it one more time to allow you to address further very minor editorial suggestions made by both reviewers and to provide code and data associated with the GLMM analysis, as suggested by Reviewer 2. Please note that it is a condition of publication in RSOS "that authors make their supporting data, code and materials available". You are welcome either to supply it as supplemental materials or to link through to some other archiving site such as Data Dryad.

We'd like to once again thank the editor and two reviewers for their careful reading and constructive comments on the manuscript.

Note that line numbers refer to the track-changed manuscript.

Data and R code for the GLMM analysis has now been deposited in DRYAD, alongside our supplementary file. These files can be accessed at:

<https://datadryad.org/stash/share/JuoPo4Xi1QEUMaO7dKGVjg6rDmre6iagsyRgaudFlpE>

Reviewer 1:

Comments to the Author(s)

Specific comments

L309: I understand that there is a lowest angle for gliding, but I don't understand what the lowest possible angle is for tailbeating even during descent. The lowest descent angle where the ODBA is significantly smaller than the level swimming? There seems a unique descent angle for each shark in Fig.S4 and this represents to be the lowest possible angle? If there is a unique descent angle for each shark, the comparison to the model should be made to see if the ascent angle of each shark is consistent with the model that minimizes the COTHD for each shark's descent angle, rather than rounding up the whole individuals.

The wording here is quite confusing. If we understand correctly, the reviewer is asking why optimal ascent and descent angles were not calculated for each individual given variability in individual descent angles.

In response, the unique descent angles do not differ significantly among individuals (as can be observed in Fig S4), and minimum descent angles approach 0° (as can also be observed in Fig S4). The variation instead comes from the angle at which sharks descend (and the proportion of the descent time they spent gliding).

In addition, this population level mean approach is required so that sufficient data can be interpreted, eliminating problems such as variable swim speeds (that could not accurately be measured as stated at line X) or any other behaviours occurring during ascent.

L319: I understood that if the ascent angle is too shallow, the time rate of ascent, which is costly, increases, which leads to an increase in COTHD. The ODBA is twice as large at an ascent angle of 0° by the regression line than at any descent angle. The increase of COTHD at the ascent angle of 0° seems to be caused by this assumption. This problem arises from the regression on the quadratic curve. The force due to gravity increases with $\sin\phi$ during ascent. In the range of 0° to 40°, the sine curve can be approximated by linear regression. This is probably because COTHD does not have a convex downward curve in the case of linear regression, but the reasoning for the regression on the quadratic curve needs to be explained. According to the author's explanation, when the ascent angle is shallow, the ascent is caused by the lift of the pectoral fin, which means that the ODBA does not increase?

We thank the reviewer for this interesting point.

We firstly argue that the sin function in this case may not be appropriate, because changes in angle might also change the configuration and production of lift, between pectoral fin, body, caudal fin. If the contribution to lift changes, the hydrodynamic efficiency changes, in turn affecting the ODBA vs angle. At very shallow angles, the vast majority of lift should come from the pectoral fins, at very steep angles, no lift needs to be produced by these fins....In short, we do not have the data to build a mechanistic model, and the quadratic is the most flexible.

Secondly, the quadratic curve fit the data better than the linear relationship (AIC: -64 versus -59), collectively indicating our data-driven approach is better.

We have now added this explanation to the methods at lines 199-203, specifically stating: 'ODBA_a was estimated from the quadratic relationship between ODBA and ϕ_a , as a quadratic model was found to fit the data better than the equivalent linear model through model testing (figure 2A; $ODBA = 2E-05 \phi_a^2 + 0.00001\phi_a + 0.0226$). This data-driven approach was selected as we did not have the data to use a mechanistic approach to test how the efficiency of lift production changed as body pitch changed.'

Reviewer 2:

Comments to the Author(s)

Thanks for addressing my previous comments on your manuscript. I can see that you have done a lot of work and I think it has noticeably improved the manuscript.

I am happy with all the changes you have made. My only remaining concern with this paper is that you have not made an attempt to make any part of your analysis reproducible. I can see that you use some non-open access software like Igor Pro and perhaps it's not easy to show the series of "clicks" involved in the data processing or analysis you carried out in that environment. However, you also use R and I feel strongly that you need to provide the code you used to fit the statistical models, since R is an open access programming language and environment. It shouldn't be difficult to do that.

You argue in your response that you don't feel it is necessary to provide code for the statistical models fitted in R because GLMMs are part of a standard framework. I would disagree, these are complex models and all of the decisions involved in fitting them are important. At the very least, I think it is important to promote good practice among readers by providing code that supports reproducibility where at all possible.

Data and R code for the GLMM analysis has now been deposited in DRYAD, alongside our supplementary file.

Given it is a condition of publication in RSOS "that authors make their supporting data, code and materials available", I would argue that this is a critical bit of presenting your work.

I have some very minor editorial suggestions. Well done for this nice piece of work.

Line 17: Double use of "patterns" is a bit repetitive, you could perhaps remove the first one.

'These patterns' have been removed at line 15.

Line 22: By low angle do you mean a small angle? If so, I would consider calling it a small angle instead.

'Lowest' has been replaced with 'smallest' at line 21.

Line 163: Do you mean models were fitted? I would use fitted when using existing software.

'Built' has been replaced with 'fitted' at line 160.

Line 171: Time series isn't hyphenated

The hyphen has been removed from time-series at line 167.